# IRRISIGHT: A Large-Scale Multimodal Dataset and Scalable Pipeline to Address Irrigation and Water Management in Agriculture

**Nibir Chandra Mandal**[*]
University of Virginia
wyr6fx@virginia.edu

**Oishee Bintey Hoque**[*]
University of Virginia
oishee30@virginia.edu

**Mandy L Wilson**
University of Virginia
alw4ey@virginia.edu

**Samarth Swarup**
University of Virginia
swarup@virginia.edu

**Sayjro K Nouwakpo**
Agricultural Research Service
United States Department of Agriculture
kossi.nouwakpo@usda.gov

**Abhijin Adiga**
University of Virginia
abhijin@virginia.edu

**Madhav Marathe**
University of Virginia
marathe@virginia.edu

## Abstract

The lack of fine-grained, large-scale datasets on water availability presents a critical barrier to applying machine learning (ML) for agricultural water management. Since there are multiple natural and anthropogenic factors that influence water availability, incorporating diverse multimodal features can significantly improve modeling performance. However, integrating such heterogeneous data is challenging due to spatial misalignments, inconsistent formats, semantic label ambiguities, and class imbalances. To address these challenges, we introduce IRRISIGHT, a large-scale, multimodal dataset spanning 20 U.S. states. It consists of 1.4 million pixel-aligned 224×224 patches that fuse satellite imagery with rich environmental attributes. We develop a robust geospatial fusion pipeline that aligns raster, vector, and point-based data on a unified 10m grid, and employ domain-informed structured prompts to convert tabular attributes into natural language. With irrigation type classification as a representative problem, the dataset is AI-ready, offering a spatially disjoint train/test split and extensive benchmarking with both vision and vision–language models. Our results demonstrate that multimodal representations substantially improve model performance, establishing a foundation for future research on water availability. https://github.com/Nibir088/IRRISIGHT https://huggingface.co/datasets/OBH30/IRRISIGHT

## 1   Introduction

Managing agricultural water use under increasing climate pressures is a growing challenge. Excessive irrigation has contributed to declining groundwater levels and reduced river discharge, posing serious threats to water security and ecosystem health [15, 79, 62, 24, 55]. A comprehensive view that integrates irrigation practices, hydrology, soil properties, and crop types is essential for understanding and managing water availability. While efficient irrigation methods can reduce water losses from

---

[*]Equal contribution.

39th Conference on Neural Information Processing Systems (NeurIPS 2025) Track on Datasets and Benchmarks.

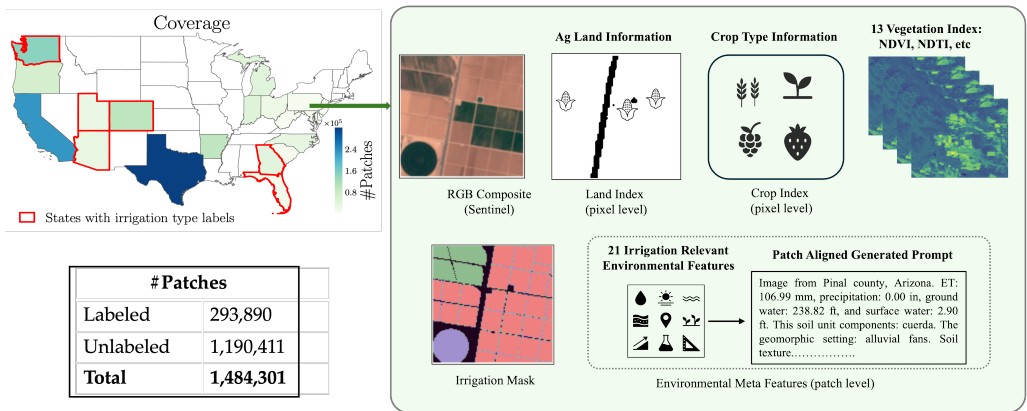

| #Patches | |
|---|---|
| Labeled | 293,890 |
| Unlabeled | 1,190,411 |
| **Total** | **1,484,301** |

Figure 1: An overview of coverage and dataset structure. (a) Shows coverage of our dataset. (b) Sample multi-modal (containing 37 meta features) ML-ready data from our dataset.

runoff—and help sustain crops during drought—they often require substantial investments from farmers [42]. Mapping water availability at high spatial resolution and across large regions is therefore critical. Such maps not only inform sustainable water management but also support climate adaptation strategies, track irrigation infrastructure investment trends, and enable regional or basin-scale modeling of the environmental impacts of shifting irrigation practices [36, 54, 3].

Remote sensing, combined with deep learning, has been extensively applied for large-scale mapping of agricultural features [35, 76, 46, 45, 12, 29, 1, 32, 23, 7, 60]. From the perspective of water availability, numerous studies have focused on distinguishing irrigated from non-irrigated lands in mixed-use agricultural areas. This has led to the creation of several prominent datasets at varying spatiotemporal scales [13, 66, 63, 40, 65, 78, 33, 11]. Another important related area is cropland monitoring [32, 31], as crop type directly influences irrigation decisions—such as the choice between sprinkler, drip, or flood systems—and therefore impacts overall water consumption. Additionally, water stress monitoring during droughts has been explored using remote sensing and deep learning approaches [75]. This work builds on our previous efforts focused on irrigation type mapping [25, 27], addressing several technical challenges identified in earlier research. A more detailed comparison with these prior studies is provided in the related works section.

The lack of fine-grained, large-scale data presents a major challenge to deploying machine learning (ML) tools for optimizing agricultural water use. Although ML has achieved success in land cover and crop classification using satellite imagery, applying these methods to water availability problems is fundamentally more complex [27]. This complexity arises from three key challenges. First, water availability is influenced by irrigation decisions as well as physical factors that depend not only on visible surface features, but also on invisible factors, including soil properties, hydrological characteristics, topography, water availability, climate patterns, and crop requirements. Second, integrating these variables from heterogeneous data sources is non-trivial due to inconsistent formats, spatial misalignment, missing data patterns, and varying temporal granularity. Aligning these onto common spatial units requires reprojection, resampling, and interpolation strategies that must preserve semantic integrity while avoiding aliasing or feature leakage. The lack of standard tools and benchmarks for this fusion process increases the risk of bias and inconsistency across regions. Third, the semantic ambiguity of class labels (in the case of irrigation type mapping for e.g., drip vs. furrow vs. sprinkler) and high class imbalances in real-world settings make classification unstable and unreliable without careful curation and domain knowledge.

**Contributions.** We present IRRISIGHT, the first large-scale, multimodal dataset designed to enable generalizable machine learning for addressing water availability problems across the United States (US). An ML-ready sample from this dataset is shown in Figure 1. Our contributions address critical gaps in data availability, representation, and evaluation by integrating diverse datasets. This is followed by extensive benchmarking using various state-of-the-art models. A summary of our contributions is as follows.

- **Novel Large-scale Multimodal Dataset:** We construct a dataset of over 1.4 million pixel-aligned 224×224 geospatial patches across twenty agriculturally important US states. Each patch integrates visual and non-visual modalities, including soil surveys, climate variables (precipitation, evapotranspiration (ET)), and hydrologic data (groundwater, surface water). To our knowledge, this is the first dataset to align such diverse modalities for water availability mapping.

- **Automated Data Fusion and Standardization Pipeline:** We develop a fully automated pipeline (see Figure 2) that integrates heterogeneous geospatial inputs—raster, vector, and point-based data—onto a common 10m grid using spatial joins, reprojection, masking, and resolution-aware resampling. Our pipeline processes 6.8TB of raw geospatial data from 11 public sources, performing standard filtering, spatial cropping, and quality control across the twenty states across the contiguous US (covering 155 million acres of agricultural land). This scalable pipeline allows us to continuously expand our geographic coverage.

- **Text-augmented Representations via Structured Prompts:** We convert non-visual features (e.g., soil texture, drainage, slope, hydrologic group) into natural language descriptions using a domain-informed, rule-based system. These prompts enable alignment with vision–language models and inject agronomic context that is not visible in imagery. Unlike one-hot or numerical feature vectors, natural language provides a semantically dense, interpretable format that supports alignment with multimodal models, facilitates human-in-the-loop analysis, and allows flexible conditioning in generative or contrastive learning settings. Thus, the text modality serves not just as auxiliary metadata, but as a linguistically structured, domain-informed signal that bridges spatial data with real-world irrigation knowledge.

- **ML-Ready Dataset and Evaluation Framework:** Using irrigation type detection as a representative problem, our work uses labeled data from six states to develop a training dataset. We provide a spatially disjoint training/testing split dataset, and an evaluation setup for states with irrigation type labels. After processing and alignment, we have a total of **1,484,301 ML-ready patches** across 20 states, with **293,890 labeled** and **1,190,206 unlabeled** examples. Spatial correlations and out-of-distribution generalizations are accounted for in the train-validate-test splitting. We also provide model-predicted labels with corresponding confidence scores for unlabeled data.

- **Extensive Benchmarking with Vision and Vision–Language Models:** We benchmark a range of models, including convolutional neural networks (CNNs), transformers, and vision-language models on our dataset for cross-state supervised learning. Our experiments demonstrate that the multimodal inputs with structured text prompts significantly improve cross-state generalization compared to image-only baselines on labeled data.

## 2   Related Work

**Existing irrigation mapping products derived by remote sensing.**   A range of remote sensing-based datasets have been created to distinguish irrigated from non-irrigated areas without providing information on specific irrigation methods. Recent efforts to map irrigation have increasingly focused on higher-resolution datasets. For instance, the AIM-HPA dataset provides 30-meter resolution irrigation maps, though it is limited to the High Plains region [11]. LANID [78] applies decision-tree methods to generate annual irrigation maps over a 20-year span, accompanied by open-access ground-truth data. IrrMapper [33] further leverages 60,000 point samples collected over 28 years, integrating Landsat imagery with climate, meteorological, and terrain features to train a Random Forest classifier. At coarser resolutions, the Moderate Resolution Irrigated Area Dataset (MIrAD) offers 250-meter resolution maps but relies heavily on census-based statistics, which can lead to considerable classification errors [6, 56]. However, none of these works address the problem of identifying irrigation types, nor do they apply state-of-the-art ML techniques.

**Related work from our team.**   This work is part of a series of works undertaken by our team in mapping irrigation infrastructure [27, 25, 26, 43]. Our recent work [27] – Knowledge-Informed Irrigation Mapping (KIIM) – takes the first steps toward integrating multimodal data for irrigation classification. It contributes methods to improve classification performance with limited labeled data and poor spatial coverage. Its precursor [25] – IrrNet – is a preliminary work undertaken where only satellite imagery is used for the purpose. Our current work is motivated by the lessons learned and challenges faced in these works. KIIM focuses on states with labeled data, and, as a result, is limited in terms of scalability due to several structural constraints. It incorporates state-specific priors using a

projection matrix, which is unavailable for most of the states. We consolidated the dataset used in KIIM and released it as IrrMap [43]. IrrMap includes labeled data for four states, along with satellite image patches, crop masks, derived indices, and land-use information. IRRISIGHT improves upon IrrMap in both geographic coverage and feature richness. It incorporates labeled data from six states and augments it with additional textual information from soil databases and hydrological sources using LLMs. This multimodal dataset is also used to benchmark a broader set of models—including visual–LLM models—beyond those evaluated in IrrMap. While IRRISIGHT builds on the foundation of IrrMap, which focused specifically on irrigation type detection, it is designed to support a wider range of water availability and management questions. In summary, while prior works such as IrrMap and KIIM laid important groundwork in irrigation mapping and modality fusion, IRRISIGHT advances the field by providing a large-scale, multimodal, and extensible benchmark designed for modern vision–language and geospatial learning (see Section C in supplement).

**Works on remote-sensing and multimodal data.** Recent work in multimodal remote sensing has introduced large-scale foundation models trained on diverse data types such as optical, SAR, and multispectral imagery to support universal earth observation tasks [22, 16] and vision-language applications like captioning [38], question answering [34, 30], and region-level reasoning [51]. Generative and contrastive frameworks further enable effective data assimilation and cross-modal representation learning [58, 39]. These advancements are complemented by emerging benchmark datasets that span modalities, spatial resolutions, and temporal dynamics, facilitating research in tasks such as segmentation [50, 18], time-series forecasting [4, 5] and multimodal text-to-image generation [41, 19]. In agriculture, these multimodal approaches improve monitoring by combining optical and multispectral data to capture crop and land use dynamics [9, 10, 47, 64].

## 3 Data Collection

Our dataset integrates geospatial data from multiple public sources, spanning raster, vector, and point-based formats. All data are harmonized to the Albers Equal Area Conic projection (EPSG:5070) for spatial consistency. Table 1 summarizes the core data sources used. Below, we provide some details about the data and its processing. More details are provided in the supplement.

**Satellite imagery:** Sentinel-2 imagery was obtained from USGS Earth Explorer. Data acquisition focused on the peak irrigation season (July), a key period for assessing water use and crop conditions, as imagery from other periods may capture snow cover, bare soil, or dormant vegetation, making it less useful for irrigation analysis. Images were filtered out if they did not meet quality criteria or were deemed redundant.

**Irrigation labels:** No national-scale irrigation label dataset is available. This data was obtained for each of the six states for which it is publicly available (See Table in Supplement). These datasets vary significantly in terms of temporal coverage, spatial granularity, and annotation standards. While all sources include field-level polygons and associated attributes, such as irrigation status, method type, crop classification, and water source, differences in data collection protocols introduce substantial label noise. For Utah, Washington, and Colorado, we have complete statewide coverage of irrigated lands. Furthermore, irrigation practices vary considerably by state. Notable class imbalances across and within states pose additional challenges for model training.

**Land use and crop type:** We utilize two national-scale raster products to identify and contextualize agricultural areas. The MRLC National Land Cover Database (NLCD) offers 30m-resolution land cover classifications across 16 categories, including cultivated cropland and pasture (See Table 1). Complementarily, the USDA Crop Data Layer (CDL) provides annual, per-pixel crop type labels from 2008 to 2023, with over 130 unique crop classes. Together, these layers are used to filter cropland regions and enrich patch-level annotations with crop-specific information.

**Water availability:** We incorporate environmental observations from the USGS National Water Information System (NWIS), which provides raw, site-specific measurements of groundwater depth, surface water elevation, and precipitation at daily temporal resolution. We preprocess this data by filtering for relevant sites within our study regions and aggregating the daily values into monthly averages spanning 2010 to 2025.

**Evapotranspiration:** We incorporate monthly evapotranspiration (ET) rasters from the USGS Famine Early Warning Systems Network (FEWS). This data provides regional-scale estimates of

water loss through soil evaporation and plant transpiration at 1km resolution. While ET is a critical factor for understanding irrigation needs, it is challenging to integrate with fine-resolution satellite imagery due to its coarse spatial granularity and atmospheric dependencies. Nonetheless, we spatially align ET values to image patches to provide a proxy for water stress and latent demand during peak growing seasons.

**Soil data:** We utilize detailed soil information from the USDA NRCS SSURGO database. The database contains a nationwide geospatial dataset compiled through extensive field surveys, lab analysis, and expert interpretation. The data are provided as vector polygons (map units), where each map unit is linked to a set of relational tables describing soil components, horizon-level measurements, texture groups, and geomorphic features. We use attributes such as slope, hydrologic group, composition percentage, organic matter, bulk density, soil texture descriptions, and geomorphic landform classifications. These properties are crucial for assessing soil water retention, permeability, and suitability for different irrigation methods.

Table 1: Summary of Data Sources Used in Dataset Construction.

| Source | Description | Spatial Resolution | Temporal Coverage | Use |
|---|---|---|---|---|
| Sentinel-2 [2] | Multispectral satellite imagery (10 bands) | 10m (20m bands upsampled) | 2014–2023 (July only) | Visual input for patch extraction |
| USGS Irrigated Lands [69] | State-level irrigation maps | 30m raster | 2002–2017 | Supervised labels for irrigation type |
| MRLC NLCD [48] | Land Use / Land Cover (LULC) classification | 30m raster | 2014–2023 | Filtering non-agricultural patches |
| USDA CDL [71] | Crop Data Layer with per-pixel crop types | 30m raster | 2008–2023 | Crop-based filtering, auxiliary input for prompts |
| USGS NWIS [72] | Groundwater, surface water, and precipitation (station-level) | Point data (lat/lon) | 2010–2025 (monthly) | Patch-level climate/hydrology features |
| USGS FEWS ET [73] | Monthly evapotranspiration rasters | 1km raster | 2014–2023 | ET features per patch |
| USDA NRCS SSURGO [52] | Soil map units, components, horizons, texture, geomorphology | 1:24,000 scale vector polygons | Static | Text prompt generation, irrigation suitability assessment |
| US Census TIGER/Line [70] | County boundary polygons with metadata (name, state, FIPS) | ~1:500,000 scale vector polygons | Static | Regional referencing, county-level context encoding |

# 4 Data Processing Pipeline

An outline of the various steps involved in data acquisition, processing, and integration is provided in Figure 2. Similarly, an outline of the preparation of the ML-ready dataset is provided in the supplement. We provide a summary of each data processing step below (detailed descriptions are provided in the supplement).

## 4.1 Data Acquisition

All datasets are summarized in Table 1. We retrieve Sentinel-2 metadata from the Copernicus Data Space Catalogue, filtering for cloud-free acquisitions within the growing season window. We remove duplicate scenes to ensure a balanced spatial coverage across the dataset. The associated satellite imagery for the selected scenes are then downloaded via authenticated API requests. Using state-level polygons, we extract overlapping land cover (from NLCD), crop type (from CDL), and soil information (SSURGO). Extracted land use and crop type rasters are reprojected to a 10m grid via nearest-neighbor interpolation. The SSURGO soil data provides polygonal soil map units with associated horizon-level and component-level attributes. For hydrological data, we queried the USGS National Water Information System to collect data for all active monitoring sites across the US states, including geolocation and site identifiers.

## 4.2 Data Processing

**Satellite imagery and other land masks.** Sentinel-2 scenes are parsed into ten multispectral bands (B02–B12) spanning 10m and 20m resolutions. These are aligned using equal-area projection and bilinear sampling. Then the stacked bands are divided into fixed-size tiles of $224 \times 224$ pixels. We retain patches only if at least 10% of pixels fall into cropland categories and contain valid reflectance

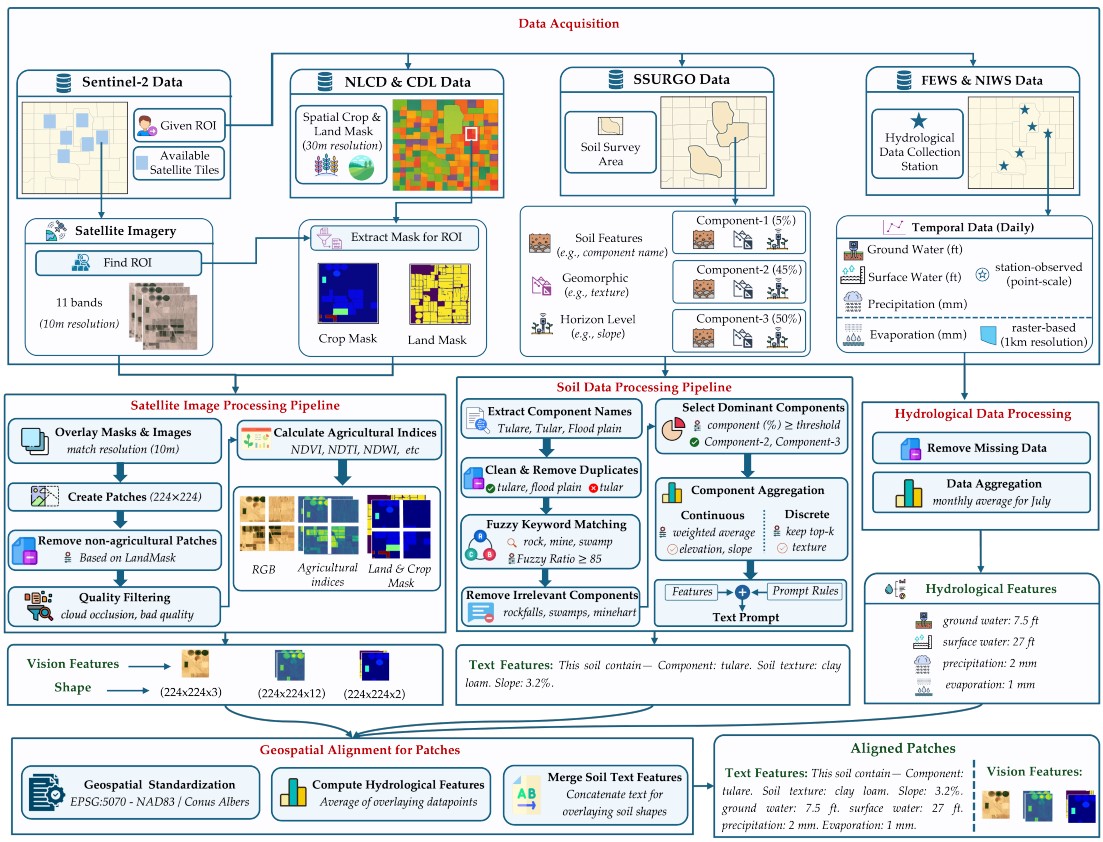

Figure 2: Multimodal Data Processing Pipeline.

values. All land use and crop masks are reprojected using nearest-neighbor interpolation to preserve categorical semantics.

**Derived indices.** To enhance surface property analysis for irrigation mapping, we compute a suite of spectral indices capturing vegetation health, water presence, and soil conditions. These include popular indices such as NDVI, GNDVI, CIgreen, NDWI, EVI, SAVI, and MSAVI. We have provided the details in the supplement.

**Soil data.** We standardize gSSURGO soil survey data to construct spatially aligned soil features suitable for ML tasks. Each mapped unit contains multiple overlapping soil components, each with its own attributes, including: (i) *horizon-level measurements* depth-specific properties such as available water capacity, organic matter, and hydraulic conductivity; (ii) *soil texture*: the relative proportions of sand, silt, and clay; and (iii) *geomorphic attributes* landscape positions such as terraces or floodplains. Each component differs in spatial dominance, which is defined by its areal proportion within the map unit. To ensure consistency, we follow a structured aggregation technique by selecting irrigation-relevant components with significant spatial coverage. The details of filtering and aggregation are in the supplement.

**Hydrological data.** For each hydrological site, we retrieved daily measurements of gauge height, groundwater level, or precipitation, and aggregated them (by taking the mean) into monthly means from 2016 to 2025. Each time series was aligned to a complete (2016–2023) year–month index to preserve temporal consistency and explicitly represent missing data.

### 4.2.1 Text-Prompt Generation

To supplement satellite imagery with non-visible but agronomically critical information not directly observable at the pixel level, we generate structured, rule-based natural language prompts that encode

localized soil and landform characteristics. These prompts enrich the feature space with domain knowledge reflecting soil and landscape attributes that influence irrigation suitability.

Each image patch is spatially joined with soil map units from the USDA NRCS **SSURGO** database. These polygons include: (i) a list of soil components (e.g., *Hanford loamy sand*) with their areal composition percentages, and (ii) attributes for each component, such as texture, drainage class, hydrologic group, slope range, geomorphic description, and horizon-level properties (e.g., bulk density, available water capacity).

For each patch, we select up to two dominant components per map unit based on areal proportion (typically >30%), representing the most influential soil types within that region. Structured attributes are then converted into coherent text using handcrafted templates. For example, attributes such as soil name (*Tulare*), drainage class (*well drained*), texture group (*sandy loam*), runoff class (*moderate*), and slope (*averages 3.2%*) are concatenated into a standardized template:

> "This soil unit contains Tulare. Soil texture includes sandy loam. It is classified as well drained with moderate runoff. The average slope is 3.2%."

If multiple soil components intersect a patch, their corresponding prompts are concatenated using a delimiter (##) to preserve provenance and reduce ambiguity. The resulting textual prompt is assigned to each image patch as a field named `text_prompt`.

## 4.3 Data Integration

To construct a unified multimodal dataset, we spatially align all heterogeneous geospatial inputs—satellite imagery, environmental variables, and soil attributes—into structured $224 \times 224$ patches. Each patch combines pixel-level reflectance with auxiliary domain features, including evapotranspiration (ET), precipitation, groundwater depth, surface water elevation, and processed soil properties. All inputs are reprojected to a common projection EPSG:5070. Patch-level values are extracted using: (i) polygonal intersection for irrigation labels and soil map units; (ii) centroid-based nearest neighbor lookup for point data (e.g., groundwater, precipitation); and (iii) raster sampling for gridded layers (e.g., ET). Soil prompts are joined to each patch via spatial intersection; if multiple soil units overlap, prompts are concatenated.

## 4.4 ML-ready Dataset Preparation

Here, we consider the representative problem of irrigation type mapping. However, our data can also be applied to related tasks by reusing the same processing pipeline provided with the dataset. After processing and alignment, we extract a total of **1,484,301 ML-ready patches** across 20 states, with **293,890 labeled** and **1,190,206 unlabeled** examples. The labeled subset comes from six states, some with partial coverage (See Figure 1), enabling semi-supervised and pretraining applications. After reprojecting, masking, and patch extraction, we obtain **5,649 processed tiles** occupying **1,386.5 GB**, which represents a **77% reduction from raw data** in storage size compared to the original data footprint.

**Training and evaluation splits.** We adopt two complementary strategies for splitting our dataset to support both intra-state generalization and cross-regional evaluation: a 70:15:15 spatial split within states, and a leave-one-state-out protocol across states. In the cross-state splitting, for each labeled state the train-validate-test split is performed at the tile level (one satellite image) as opposed to splitting the set of individual patches. It ensures that data from the same geographic region does not appear in multiple splits—thereby preventing spatial leakage. This design is critical in remote sensing settings, where nearby patches are highly correlated due to shared land cover and climate conditions. To evaluate out-of-distribution generalization, we perform a leave-one-state-out split. A target state is held out entirely for testing, while the remaining states contribute data to the training and validation sets. For the training states, we again split at the tile level and reserve a small portion (typically 10%) for validation. This setting simulates real-world scenarios where labeled data is unavailable in a new geographic region, and the model must transfer knowledge learned from other states. The details are in the supplement.

# 5 Dataset Benchmarking

We evaluate segmentation performance across three irrigation types (Flood, Sprinkler, and Drip) using nine baseline models and multiple modalities using standard Dice and IoU metrics (details in the supplement). Table 2 reports overall metrics, while Table 3 details per-state Dice scores under the leave-one-state-out setting.

**Architecture Comparison.** Among RGB-only architectures, transformer-based models (e.g., Seg-Former, FarSeg) outperform CNN baselines. However, performance improvements from architectural changes are modest compared to those achieved by integrating additional modalities. Vision-language models like RemoteCLIP leverage text supervision to improve generalization, while KIIM benefits from structured auxiliary inputs (crop and land use). KIIM (RGB + Crop + Land) achieves the highest performance across all types, with Dice scores of 93.6% (Flood), 95.8% (Sprinkler), and 94.6% (Drip). RemoteCLIP and CLIP, which incorporate textual prompts, also surpass all RGB-only models. This indicates the utility of language-grounded supervision. These results underscore that incorporating domain-specific context is more impactful than architecture alone for the irrigation mapping task.

**Cross-State Generalization.** KIIM maintains strong performance across diverse states, likely due to its explicit fusion of structured crop and land metadata with visual features, while RGB-only models degrade significantly in out-of-distribution regions. CLIP and RemoteCLIP perform well in states with clearer visual-text alignment (e.g., AZ, CO) but struggle in regions with less distinctive irrigation patterns (e.g., WA, FL), which highlights the limits of vision-only and VLM-based generalization without structured context. Among the irrigation types, sprinkler irrigation is consistently the easiest to segment (due to its circular geometric shape), with most models achieving Dice greater than 85%. Flood irrigation shows variability across states. Drip irrigation is the most challenging, particularly in CO and UT, where even strong models like KIIM score below 15%, likely due to its irregular shapes and fine-grained, low-contrast footprint.

Table 2: Performance (%) across irrigation types using different models and modalities.

| Model | Modality | Flood | | Sprinkler | | Drip | |
|---|---|---|---|---|---|---|---|
| | | Dice | IoU | Dice | IoU | Dice | IoU |
| ResNet | RGB | 35.2 | 21.4 | 92.2 | 85.5 | 88.5 | 79.4 |
| ViT | RGB | 82.9 | 70.9 | 89.9 | 81.7 | 84.1 | 72.6 |
| FPN | RGB | 86.2 | 75.7 | 91.4 | 84.1 | 86.4 | 76.0 |
| SegFormer | RGB | 86.2 | 75.8 | 91.7 | 84.7 | 85.9 | 75.3 |
| DeepLabV3+ | RGB | 87.2 | 77.4 | 92.1 | 85.4 | 86.9 | 76.8 |
| FarSeg | RGB | 88.2 | 78.9 | 92.3 | 85.7 | 89.2 | 80.5 |
| CLIP | RGB+Text | 90.1 | 82.0 | 93.1 | 87.0 | 90.7 | 83.0 |
| RemoteCLIP | RGB+Text | 90.9 | 83.3 | 93.7 | 88.2 | 92.3 | 85.7 |
| KIIM | RGB+Land+Crop | 93.6 | 88.0 | 95.8 | 91.9 | 94.6 | 89.7 |

**Impact of Textual Prompts.** As shown in Table 2, incorporating these text prompts substantially improves model performance compared to RGB-only baselines. In RemoteCLIP, the textual prompt serves as an independent semantic input encoded via a language encoder, complementing the visual encoder rather than being concatenated as auxiliary features. For instance, RemoteCLIP achieves a 92.3% Dice score on drip irrigation, compared to 88.5% for an RGB-only ResNet, confirming that structured domain knowledge enhances model discrimination across all irrigation types.

**Label Generation for Unlabeled States.** Following the performance of KIIM model, we generate synthetic labels for 17 unlabeled states. We trained the KIIM model on all six labeled data and generated irrigation maps for the unlabeled states. We show the confidence for synthetic labels in Figure 3 for seven states.

# 6 Application of the Dataset

The IRRISIGHT dataset is designed to support a range of machine learning and agricultural modeling tasks. A direct application is that it lays a pipeline for ingesting feature-rich ML-ready multimodal dataset that can help address at the national-scale various questions in the context of water availability.

Table 3: Dice scores (%) for Flood, Sprinkler, and Drip irrigation types across 6 U.S. states and 9 model architectures. Flood is not applicable in Georgia.

| State | Type | KIIM | R-CLIP | ViT | FarSeg | CLIP | DLV3+ | FPN | SegFormer | ResNet |
|---|---|---|---|---|---|---|---|---|---|---|
| AZ | Flood | **74.6** | 67.0 | 72.1 | 70.6 | 66.7 | 70.69 | 71.33 | 70.07 | 70.88 |
| | Sprinkler | 83.2 | 82.8 | **84.5** | 82.8 | 79.7 | 83.01 | 82.43 | 85.62 | 83.31 |
| | Drip | 22.5 | 5.8 | 10.3 | **29.4** | 1.8 | 18.11 | 21.92 | 23.04 | 17.54 |
| CO | Flood | 52.8 | 12.1 | **53.1** | 52.2 | 44.1 | 50.31 | 49.44 | 46.98 | 50.52 |
| | Sprinkler | **79.0** | 73.4 | 76.1 | 78.6 | 73.5 | 76.53 | 77.84 | 78.27 | 76.29 |
| | Drip | **1.9** | 0.6 | 0.5 | 1.1 | 0.1 | 0.76 | 0.67 | 1.27 | 0.19 |
| UT | Flood | **61.6** | 28.1 | 60.6 | 60.3 | 54.9 | 59.60 | 57.98 | 60.09 | 59.71 |
| | Sprinkler | **67.9** | 56.4 | 61.5 | 65.3 | 57.5 | 62.97 | 62.59 | 63.68 | 63.55 |
| | Drip | **14.8** | 0.0 | 6.4 | 0.8 | 0.0 | 9.73 | 4.23 | 4.12 | 6.46 |
| WA | Flood | **29.5** | 19.7 | 21.6 | 25.6 | 18.3 | 19.91 | 22.83 | 23.65 | 22.23 |
| | Sprinkler | **79.7** | 74.3 | 77.4 | 78.6 | 73.3 | 76.66 | 77.38 | 77.74 | 77.09 |
| | Drip | 8.9 | 1.6 | **22.1** | 12.7 | 4.5 | 18.25 | 10.05 | 27.18 | 0.01 |
| FL | Flood | 11.6 | 3.0 | 0.8 | 17.3 | 2.2 | 1.60 | 8.46 | 5.11 | **17.91** |
| | Sprinkler | **64.8** | 51.0 | 54.5 | 61.0 | 50.6 | 59.76 | 59.58 | 57.57 | 61.14 |
| | Drip | 15.2 | 2.7 | **30.3** | 28.2 | 8.5 | 6.14 | 11.51 | 8.08 | 3.70 |
| GE | Sprinkler | **59.0** | 42.4 | 48.9 | 50.4 | 37.2 | 27.12 | 47.33 | 45.93 | 47.38 |
| | Drip | 6.2 | 1.6 | 5.5 | **6.8** | 3.3 | 1.55 | 6.18 | 11.34 | 2.95 |

While irrigation type segmentation is the primary benchmark, *IRRISIGHT* was designed for broader agricultural water management tasks. Its modular multimodal pipeline—integrating Sentinel-2 imagery with soil, hydrological, and crop metadata—enables generalization beyond irrigation mapping. To illustrate this extensibility, we evaluated two downstream tasks: *crop classification* and *environmental variable regression*. The KIIM model retrained on IRRISIGHT data significantly outperformed the USDA CropScape [49], achieving higher Macro-F1 scores across all states (Supplement Table S10). Regression experiments using tree-based models also showed strong predictive accuracy for evapotranspiration, groundwater, precipitation, and surface water (Table 4), demonstrating IRRISIGHT's utility across diverse hydrological prediction tasks.

Furthermore, regression experiments using Random Forest, Gradient Boosting, and XGBoost models were conducted to predict key environmental variables—evapotranspiration (ET), groundwater, precipitation, and surface water—using other available features such as irrigation labels and geospatial attributes.. All models achieved strong coefficients of determination ($R^2$ ranging from 0.43 to 0.99) across variables (Table 4), confirming that the dataset's integrated structure enables robust learning for hydrologically relevant targets.

Table 4: Regression performance (MAE, RMSE, and $R^2$) for environmental variables using tree-based models. Lower MAE/RMSE and higher $R^2$ are better.

| Variable (Unit) | Model | MAE | RMSE | $R^2$ |
|---|---|---|---|---|
| ET (mm) | Random Forest | 22.993 | 30.279 | 0.483 |
| | Gradient Boosting | 24.751 | 31.705 | 0.433 |
| | XGBoost | 23.075 | 30.328 | 0.481 |
| Ground Water (ft) | Random Forest | 3.230 | 19.544 | 0.867 |
| | Gradient Boosting | 16.314 | 36.546 | 0.536 |
| | XGBoost | 10.683 | 28.671 | 0.715 |
| Precipitation (in) | Random Forest | 0.001 | 0.005 | 0.988 |
| | Gradient Boosting | 0.009 | 0.016 | 0.869 |
| | XGBoost | 0.003 | 0.008 | 0.963 |
| Surface Water (ft) | Random Forest | 5.782 | 64.259 | 0.986 |
| | Gradient Boosting | 60.448 | 174.265 | 0.901 |
| | XGBoost | 27.827 | 125.645 | 0.948 |

These findings provide strong empirical evidence that *IRRISIGHT* generalizes beyond irrigation classification, supporting diverse supervised and regression-based tasks in agricultural water manage-

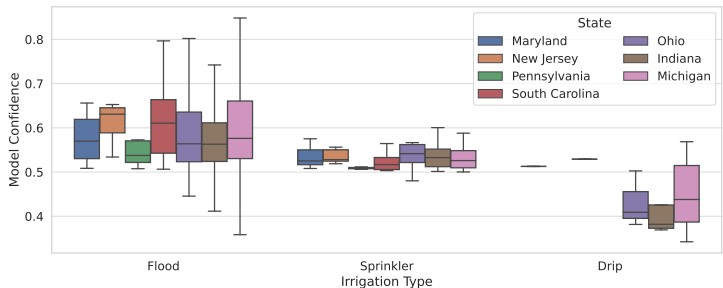

Figure 3: Distribution of model confidence scores across irrigation types (excluding non-irrigated) for the top 7 high-confidence states. Each boxplot shows the variation in predicted confidence across patches within a given irrigation class and state. Confidence values are derived from the model's softmax outputs, and higher scores indicate greater certainty in the predicted irrigation type.

ment. By design, it enables research in drought assessment, fallowing detection, irrigation suitability mapping, and crop–irrigation optimization. Some other example applications are provided below.

**Irrigation Classification with Supervised and Semi-Supervised Learning:** The labeled subset enables training of supervised deep learning models for irrigation detection, while the large unlabeled regions support semi-supervised approaches that leverage weak labels and spatial context.

**Flash Drought Detection:** A flash drought is a sudden onset drought that lasts long enough to impact vegetation [37]. This dataset can be combined with flash drought data [e.g., 53] to develop models of flash drought impact on crops and which irrigation practices are best at minimizing it.

**Fallowing Detection and Prediction:** Fallowing, where a field is not planted for a season, can be done as part of a normal crop rotation sequence, or due to lack of water availability (drought), or as part of a strategy for long-term water security [59, 57]. The dataset can be used to develop fallowing detection and prediction models, which can in turn be used to discover normal crop rotation patterns and to quantify the impact of drought.

**Irrigation Suitability Mapping:** By combining ET, precipitation, and groundwater data, the dataset can be used to estimate irrigation feasibility in areas with limited infrastructure or policy-relevant water stress.

**Region-Specific Analysis for Crop Planning:** Crop type labels, where available, enable downstream tasks such as identifying crop-irrigation suitability patterns and generating region-specific irrigation recommendations based on soil and water constraints.

# 7   Limitations

Here, we acknowledge some of the shortcomings of the IRRISIGHT datasets and describe possible future directions. A major limitation is that pixel-level labels are included for only six states. Further, not all states have full coverage. While KIIM [27] shows superior performance in irrigation type mapping, the poor results on cross-state generalization suggest the importance of providing the model with features and if possible sample labels corresponding to the target region for training to improve performance. Also, for states where limited or no label information is present, validation becomes a challenge. Using data such as US Agricultural Census, an aggregate-level validation based on county data is a possibility but has its own challenges. Raster reprojection to a 10m grid (e.g., crop type, land use) may cause misalignment or information loss, especially in heterogeneous landscapes. Our soil aggregation emphasizes dominant components, potentially removing hydrologically relevant minority types. In the future, we will explore rigorous evaluation strategies for unlabeled states, including census-model alignment and expert review.

## Acknowledgments

This material is based upon work supported by the AI Research Institutes program supported by NSF and USDA-NIFA under the AI Institute: Agricultural AI for Transforming Workforce and Decision Support (AgAID) award No. 2021-67021-35344. This work was partially supported by University of Virginia Strategic Investment Fund award number SIF160.

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
