# Supplementary Information:
# IRRISIGHT: A Large-Scale Multimodal Dataset and Benchmark for Irrigation Mapping from Satellite Imagery and Structured Environmental Features

## A    Data Collection (Additional Details for Section 3)

Our dataset integrates geospatial data from multiple public sources, spanning raster, vector, and point-based formats. All data are harmonized to the Albers Equal Area Conic projection (EPSG:5070) for spatial consistency. Table 1 summarizes the core data sources used. Below, we describe each modality and its metadata in more detail.

Table S1: State-wise Farms and Irrigation Coverage Summary for the labeled dataset.

| State | Source | Farms | Total (acres) | Irrigation Type (%) | | | Year Range | Spatial Coverage |
|-------|--------|-------|---------------|---------------------|--|--|------------|------------------|
| | | | | Sprinkler | Drip | Flood | | |
| WA | WSDA [77] | 12,696 | 2,000,402 | 84.9% | 7.1% | 8.0% | 2016-20 | 100% |
| UT | WRLU [74] | 11,404 | 1,776,424 | 53.3% | 0.1% | 46.1% | 2023 | 100% |
| CO | CDSS [8] | 15,203 | 2,534,391 | 40.7% | 0.2% | 59.2% | 2016-20 | 100% |
| AZ | USGS-VIAgL [69] | 4,701 | 222,135 | 13.0% | 43.1% | 31.0% | 2016-17 | 24.7% |
| FL | USGS-VIAgL [69] | 11,991 | 554,587 | 51.0% | 21.7% | 25.5% | 2016-17 | 37.5% |
| GA | USGS-VIAgL [69] | 6,391 | 997,630 | 5.3% | 94.7% | - | 2016 | 77.5% |

**Satellite imagery:** Sentinel-2 imagery were obtained from USGS Earth Explorer for 20 studied states of the US, covering their respective study periods. Sentinel-2A offers 10m visible and near-infrared resolution with a 5-day revisit cycle. Data acquisition focused on the peak irrigation season (July), a key period for assessing water use and crop conditions, as imagery from other periods may capture snow cover, bare soil, or dormant vegetation, making it less useful for irrigation analysis. Images exceeding 5% cloud cover, snow, or poor quality—identified via the Quality Assessment (QA) band—were excluded. The imagery includes ten spectral bands: four at 10m resolution (B02–B04, B08) and six at 20m resolution (B05–B07, B11–B12). Moreover, we excluded spatially redundant satellite images.

**Irrigation labels:** Since *irrigation label data* is not uniformly available from a single national source, we curated irrigation annotations for six US states by integrating diverse regional datasets. For Utah (2023), we utilized the Water-Related Land Use (WRLU) dataset[2], while irrigation maps for Washington (2015–2020) were obtained from the Washington State Department of Agriculture Agricultural Land Use dataset (WSDA)[3]. For Colorado (2018–2020), we incorporated shapefiles from the Colorado Decision Support System (CDSS)[4], which provides GIS layers for agricultural parcels across river basins. For six additional states (i.e., Arizona, Florida, Georgia, Missouri, New Mexico, and Texas), we adopted the USGS Verified Irrigated Agricultural Lands dataset[5], a geodatabase produced by the U.S. Geological Survey in collaboration with the University of Wisconsin. These datasets vary significantly in terms of temporal coverage, spatial granularity, and annotation standards. While all sources include field-level polygons and associated attributes, such as irrigation status, method type (e.g., drip, pivot, flood), crop classification, and water source, differences in data collection protocols introduce substantial label noise. We summarize the coverage and content of each state's dataset in Table S1. Notably, for Utah, Washington, and Colorado, we obtained complete statewide coverage of irrigated lands. In contrast, the data for Arizona, Florida, Georgia, Missouri, New Mexico, and Texas only cover selected counties or specific agricultural regions. Furthermore, irrigation practices vary considerably by state. For example, Washington and Utah demonstrate

---

[2] Water-Related Land Use, Utah

[3] Agricultural Land Use, Washington

[4] Division of Water Resource, Colorado

[5] Verified Irrigated Agricultural Lands dataset (USDA, 2002–2017)

high adoption of sprinkler-based systems, while Colorado relies predominantly on flood irrigation, a pattern often dictated by terrain and legacy water rights. These disparities in irrigation methods are also reflected in their statistical distributions: Utah's 1.76 million acres include only 0.01% drip irrigation, while Arizona's 412,263 acres include 12% under drip systems. This heterogeneity results in notable class imbalance across and within states, which poses additional challenges for model training. Such imbalance, coupled with noisy labels and partial coverage, underscores the importance of our dataset's unified representation, standardized structure, and inclusion of auxiliary modalities to support robust learning under weak supervision.

**Land use and crop type:** We utilize two national-scale raster products to identify and contextualize agricultural areas. The MRLC National Land Cover Database (NLCD)[6] offers 30m-resolution land cover classifications across 16 categories, including cultivated cropland and pasture. Complementarily, the USDA Crop Data Layer (CDL) [7] provides annual, per-pixel crop type labels from 2008 to 2023, with over 130 unique crop classes. This crop class includes corn, soybeans, cotton, alfalfa, orchards, etc. Together, these layers are used to filter cropland regions and enrich patch-level annotations with crop-specific information.

**Water availability:** We incorporate environmental observations from the USGS National Water Information System (NWIS)[8], which provides raw, site-specific measurements of groundwater depth, surface water elevation, and precipitation at daily temporal resolution. These measurements are recorded at fixed monitoring stations across the United States, with each entry associated with geographic coordinates. We preprocess this data by filtering for relevant sites within our study regions and aggregating the daily values into monthly averages spanning 2010 to 2025.

**Evapotranspiration:** To estimate crop water demand, we incorporate monthly evapotranspiration (ET) rasters from the USGS Famine Early Warning Systems Network (FEWS) [9]. These data provide regional-scale estimates of water loss through soil evaporation and plant transpiration at 1km resolution. While ET is a critical factor for understanding irrigation needs, it is challenging to integrate with fine-resolution satellite imagery due to its coarse spatial granularity and atmospheric dependencies. Nonetheless, we spatially align ET values to image patches to provide a proxy for water stress and latent demand during peak growing season.

**Soil data:** We utilize detailed soil information from the USDA NRCS SSURGO database [10]. The database contains a nationwide geospatial dataset compiled through extensive field surveys, lab analysis, and expert interpretation. Note that the data are provided as 1:24,000-scale vector polygons (map units), where each map unit is linked to a set of relational tables describing soil components, horizon-level measurements, texture groups, and geomorphic features. Specifically, we use attributes from the `component` table (e.g., slope, hydrologic group, drainage class, irrigation capability, composition percentage), the `chorizon` table (e.g., available water capacity, saturated hydraulic conductivity, organic matter, bulk density), the `chtexturegrp` table (soil texture descriptions), and the `cogeomordesc` table (geomorphic landform classifications). These properties are crucial for assessing soil water retention, permeability, and suitability for different irrigation methods. However, several challenges arise from the hierarchical and compositional nature of the data. Each polygon may contain multiple components, each with its own set of horizons and weighted percentages, requiring careful aggregation. Additionally, terminology varies across states and years, and some fields (e.g., irrigation capability) are sparse or inconsistently coded. Despite these limitations, soil attributes provide valuable non-visual context for understanding irrigation behavior, especially in cases where visual signals in satellite imagery are ambiguous.

---

[6] https://www.mrlc.gov/

[7] https://nassgeodata.gmu.edu/CropScape/

[8] https://waterdata.usgs.gov/nwis

[9] https://earlywarning.usgs.gov/fews/product/460/

[10] https://nrcs.app.box.com/v/soils/folder/233398887779

**County boundaries:** We use administrative boundary data from the U.S. Census Bureau's TIGER/Line dataset[11], which provides vector geometries for all U.S. counties. Each polygon includes metadata such as county name, state name, and FIPS codes. This dataset enables spatial reference and regional context for downstream statistical analysis and cross-county comparisons.

# B    Data Processing Pipeline (Additional Details for Section 4)

An outline of the various steps involved in data acquisition, processing and integration is provided in Figure 2. Similarly, an outline of the preparation of the ML-ready dataset is provided in Figure S1.

## B.1    Data Acquisition

### B.1.1    Sentinel-2 Data Acquisition

We retrieve Sentinel-2 metadata from the Copernicus Data Space Catalogue, filtering for cloud-free (less than 5% cloud cover) acquisitions within the growing season window (i.e., June 30 to August 1) by using state-level polygon boundaries. To reduce redundancy and avoid over-representation of frequently imaged areas, we remove duplicate scenes (i.e., multiple acquisitions covering the same spatial footprint) to ensure a balanced spatial coverage across the dataset. The associated sentinel-2 image products for the selected scenes are then downloaded via authenticated API requests.

### B.1.2    Crop, Land-Use & Soil Data Acquisition

Using state-level polygons, we extract overlapping land cover (from NLCD), crop type (from CDL), and soil information (from SSURGO). Extracted land use and crop type rasters are reprojected to a 10m grid via nearest-neighbor interpolation. The SSURGO soil data provides polygonal soil map units with associated horizon-level and component-level attributes.

### B.1.3    Hydrological Data Acquisition

We queried the USGS National Water Information System to collect metadata for all active monitoring sites across U.S. states, including geolocation and site identifiers.

## B.2    Data Processing

### B.2.1    Satellite Image Processing

Sentinel-2 scenes are parsed into ten multispectral bands (B02–B12) spanning 10m and 20m resolutions. Each band is reprojected to a common target projection (EPSG:5070), which preserves area across the continental U.S., using bilinear resampling. We define a canonical transform (i.e., a standardized spatial reference system and grid layout) and output grid based on the first valid band and resample all others into this shared space to ensure sub-pixel alignment across bands. Then the stacked bands are divided into fixed-size tiles of $224 \times 224$ pixels. We retain patches only if they meet the following criteria:

- **Cropland threshold:** At least 10% of pixels must fall into cropland categories (class 81/82) from the USDA Cropland Data Layer.

- **Valid reflectance:** More than 80% of the patch must contain valid (non-zero) reflectance values.

These filters ensure that selected patches are agriculturally meaningful and spatially coherent. All land use and crop masks are reprojected using nearest-neighbor interpolation to preserve categorical semantics. The state-wise count of tiles and patches is provided in Table S2.

---

[11]https://www.census.gov/geographies/mapping-files/time-series/geo/tiger-line-file.html

Table S2: Summary of Collected and Processed Data by State and Label Status

| Status | State | Collected | | Processed | | |
|---|---|---|---|---|---|---|
| | | #Tiles | Storage (GB) | #Tiles | Storage (GB) | #Patches |
| **Labeled** | Washington | 3428 | 813 | 1029 | 181.0 | 145336 |
| | Utah | 1030 | 253 | 279 | 31.0 | 24847 |
| | Colorado | 5411 | 1224 | 717 | 109.5 | 97087 |
| | Arizona | 3053 | 736 | 205 | 11.8 | 9560 |
| | Florida | 303 | 92 | 24 | 3.2 | 2463 |
| | Georgia | 292 | 94 | 50 | 16.0 | 12597 |
| | **Total (Labeled)** | **15517** | **3212** | **2304** | **352.5** | **293890** |
| **Unlabeled** | Arizona | 1794 | 446 | 367 | 20.0 | 18811 |
| | Florida | 265 | 84 | 69 | 11.0 | 9681 |
| | Georgia | 127 | 42 | 67 | 26.0 | 24887 |
| | Texas | 1800 | 636 | 796 | 331.0 | 318740 |
| | Arkansas | 244 | 155 | 240 | 99.0 | 95361 |
| | California | 2022 | 1200 | 1311 | 238.0 | 229002 |
| | Nebraska | 1640 | 305 | 613 | 317.0 | 305115 |
| | Ohio | 84 | 28 | 51 | 20 | 18608 |
| | North Carolina | 219 | 64 | 130 | 32.0 | 30765 |
| | South Carolina | 93 | 32 | 32 | 7.1 | 6750 |
| | Pennsylvania | 64 | 21 | 39 | 4.8 | 4588 |
| | West Virginia | 31 | 17 | 26 | 1.8 | 1721 |
| | New Jersey | 56 | 15 | 24 | 1.7 | 1619 |
| | Michigan | 341 | 101 | 199 | 44.0 | 42086 |
| | Indiana | 122 | 35 | 52 | 47.0 | 44371 |
| | Maryland | 57 | 19 | 13 | 2.1 | 1969 |
| | Oregon | 1640 | 305 | 430 | 68 | 65437 |
| | **Total (Unlabeled)** | **11,569** | **3,595** | **4,489** | **1,290.5** | **1,190,411** |
| **Total (All)** | | **27086** | **6807** | **6793** | **1643** | **1,484,301** |

To enhance surface property analysis for irrigation mapping, we compute a suite of spectral indices capturing vegetation health, water presence, and soil conditions[12]. Common vegetation indices such as NDVI, GNDVI, and CIgreen quantify canopy vigor and chlorophyll content, while EVI, SAVI, and MSAVI account for atmospheric and soil background effects [44, 68, 28]. NDWI estimates water content and drought stress [17], and NDTI captures tillage status [80]. Additional indices such as PRI, OSAVI, and WDRVI further enrich vegetation stress detection and biomass sensitivity [20, 14, 67], while RVI offers a robust alternative to NDVI under varying conditions [21]. These derived bands are appended to the spectral stack, providing physically interpretable features that complement raw reflectance and aid in downstream irrigation classification. (See Appendix Table S11 for full index definitions.)

### B.2.2 Soil Data Processing

We standardize gSSURGO soil survey data from the USDA NRCS to construct spatially aligned soil features suitable for machine learning tasks. This database has a hierarchical structure: each mapped unit contains multiple overlapping soil components, each with its own attributes, including (i) *horizon-level measurements* (depth-specific properties such as available water capacity, organic matter, and hydraulic conductivity); (ii) *soil texture* (the relative proportions of sand, silt, and clay); and (iii) *geomorphic attributes* (landscape positions such as terraces or floodplains). Each component differs in spatial dominance, which is defined by its areal proportion within the map unit. To ensure consistency, we follow a structured aggregation technique:

---

[12]https://www.nv5geospatialsoftware.com/docs/AlphabeticalListSpectralIndices.html

- **Irrigability filtering**: We systematically exclude soil components and geomorphic features unsuitable for irrigation using rule-based fuzzy keyword matching against curated exclusion lists. Component names and geomorphic feature names are extracted from the dataset, cleaned, deduplicated, and compared to expert-defined keywords representing non-irrigable conditions. For instance, irrigation can not happen in rocky substrates, highly permeable sands, saline soils, saturated wetlands, and unstable slopes. We apply partial string similarity thresholds (fuzzy ratio $\geq$ 85 for components and geomorphic features) to ensure robust matching despite naming inconsistencies. We include all exclusion lists in Table S6. This process filtered 1120 out of 14928 total soil components (7.5%) and 65 out of 404 geomorphic features (16.1%) as irrigation-unsuitable. The goal of this filtering is to retain only components and landforms that are physically relevant to irrigation. Therefore, it improves the semantic coherence and practical utility of the resulting soil attributes. This contributes a physically grounded (i.e., based on real and observable physical properties of the system) and semantically aligned preprocessing step that supports the extraction of irrigation-relevant insights. Moreover, this ensures that retained components reflect realistic and agriculturally meaningful soil and landform conditions.

- **Dominant component selection**: We retain components with above-median areal composition to focus on physically representative contributors. We use the median threshold to robustly filter minor inclusions and reduce sensitivity to small, fragmented components.

- **Feature aggregation**: Continuous variables (e.g., slope, elevation, water capacity) are aggregated using composition-weighted averages. Categorical variables (e.g., hydrologic group, drainage class) are resolved based on the most dominant component, while descriptive categorical fields such as soil texture and component names are preserved by retaining the top-ranked values according to areal proportion.

This standardization transforms multi-resolution soil data into uniform and interpretable (i.e., aligned with domain concepts used in agronomy and hydrology, e.g., slope %, drainage class, texture categories) spatial features that preserve relevant physical variability while reducing noise.

### B.2.3 Hydrological Data Processing

For each hydrological site, retrieved daily measurements of guage height (i.e., the water surface elevation above a predefined reference point, measured in feet), groundwater level (measured in feet), or precipitation (measured in inches) and aggregated (by taking the mean) them into monthly means from 2016 to 2025. Each time series was aligned to a complete (2016–2023) year–month index to preserve temporal consistency and explicitly represent missing data.

### B.2.4 Text-Prompt-Generation

To supplement satellite imagery with non-visible but agronomically critical information, we generate structured natural language prompts that describe localized soil and landform characteristics. Features, such as soil component names, hydrologic group, irrigation capability class, etc., are not directly observable from remote sensing. However, these features are essential for human decision-making in irrigation and land management. For example, high runoff potential, poor drainage, or rocky geomorphology may preclude irrigation despite vegetation signals in satellite data. This textual representation also encodes the local geospatial context implicitly through place-specific components and geomorphic formations (e.g., *alluvial fan, panoche soil, playa basin*), which may reflect historical sedimentation, erosion, or aquifer accessibility, factors critical for sustainable irrigation planning. Unlike one-hot or numerical feature vectors, natural language provides a semantically dense, interpretable format that supports alignment with multimodal models, facilitates human-in-the-loop analysis, and allows flexible conditioning in generative or contrastive learning settings. Thus, the text modality serves not just as auxiliary metadata but as a linguistically structured, domain-informed signal that bridges spatial data with real-world irrigation knowledge. We show the rules for prompt generation in Table S4.

Table S3: Standardization of Raw Irrigation Labels into irrigation class and subclasses.

| Original Label | Mapped SubClass | Mapped Irrigation |
|---|---|---|
| Drip/Micro-Sprinkler, Micro, Micro Sprinkler, Micro-Bubbler, Micro-Drip, Micro-Sprinkler, Mirco-Bubbler | Micro-Drip | Drip |
| Big Gun/Drip, DRIP, Drip, Drip Microirrigation, Drip/Big Gun, Drip/None, Drip/Rill, Drip/Wheel Line | Drip | |
| FLOOD, FURROW, Flood, GATED_PIPE, Hand/Rill, None/Rill, Rill | Flood | Flood |
| Center Pivot, Center Pivot/Drip/Sprinkler, Center Pivot/None, Center Pivot/Rill/Wheel Line, Center Pivot/Sprinkler, Center Pivot/Sprinkler/Wheel Line, Center Pivot/Wheel Line | Center Pivot | Sprinkler |
| Big Gun, Big Gun/Center Pivot, Big Gun/Sprinkler, Big Gun/Wheel Line, Traveling Gun, Traveling Gun, pivot, sprinkler | Big-Gun | |
| Center Pivot - Tow, Hand/Sprinkler, Lateral Sprinkler, Other Sprinkler, Overhead, Rill/Sprinkler/Wheel Line, SPRINKLER, Side Roll, Solid State Sprinkler, Sprinkler, Sprinkler & Bubbler, Sprinkler/Wheel Line, Traveler Sprinkler, Wheel Line, Wheel line | Sprinkler | |
| Big Gun/Drip, Center Pivot/Drip/Sprinkler, Drip/Big Gun, Drip/Micro-Sprinkler, Drip/None, Drip/Rill, Drip/Rill/Sprinkler, Drip/Sprinkler, Drip/Sprinkler/Wheel Line, Drip/Wheel Line, Dry Crop, Micro, None/Sprinkler, None/Sprinkler/Wheel Line, None/Wheel Line, Non_irrigated, NON_IRRIGATED, Not Specified, Rill/Sprinkler, Rill/Sprinkler/Wheel Line, Solid State Sprinkler, Drip Microirrigation, Sprinkler And Drip, Sub-irrigated, UNKNOWN, Unknown, Uncertain | Removed | Removed |

## B.3   Data Integration

To construct a unified multimodal dataset, we spatially align all heterogeneous geospatial inputs—satellite imagery, environmental variables, and soil attributes—into structured $224 \times 224$ patches. Each patch combines pixel-level reflectance with auxiliary domain features, including July-aggregated evapotranspiration (ET), precipitation, groundwater depth, surface water elevation, and processed soil properties. County-level boundaries are also assigned to provide administrative context.

All inputs are reprojected to a common EPSG:5070 grid. Patch-level values are extracted using ($i$)polygonal intersection for irrigation labels and soil map units, ($ii$) centroid-based nearest neighbor lookup for point data (e.g., groundwater, precipitation), and ($iii$) raster sampling for gridded layers (e.g., ET). Soil prompts are joined to each patch via spatial intersection; if multiple soil units overlap, prompts are concatenated. Missing prompts are explicitly flagged.

Table S4: Field-wise Prompt Generation Rules and Template

| Field | Prompt Template (Rule) |
|---|---|
| Component Names | This soil unit contains the following dominant components: `<compname-1>`; ...;`<compname-5>`. |
| Geomorphic Features | The geomorphic setting includes: `<geomorphic-1>`;...;`<geomorphic-5>`. |
| Texture Classes | Soil texture: `<texture-1>`;...;`<texture-5>`. |
| Runoff and Drainage | The soil has a runoff class of `<runoff>` and drainage class `<drainagecl>`. |
| Hydrologic Group | Hydrologic group: `<hydgrp>`. |
| Hydric Rating | It is rated as `<hydric>`. |
| Irrigation Capability | Irrigation capability: `<irrcapcl>`, subclass `<irrcapscl>`. |
| Slope and Elevation | Average slope: `<slope>`%, elevation: `<elevation>` m. |
| Soil Properties | Soil properties: AWC = `<awc>`, Ksat = `<ksat>`, OM = `<om>`, BD = `<dbovendry>`, water content at 1/10 = `<wtenthbar>`, at 15 bar = `<wfifteenbar>`. |
| **Final Prompt Structure** | This soil unit contains the following dominant components: `<compname-1>`; ...; `<compname-5>`. The geomorphic setting includes: `<geomorphic-1>`; ...; `<geomorphic-5>`. Soil texture: `<texture-1>`; ...; `<texture-5>`. The soil has a runoff class of `<runoff>` and drainage class `<drainagecl>`. Hydrologic group: `<hydgrp>`. It is rated as `<hydric>`. Irrigation capability: `<irrcapcl>`, subclass `<irrcapscl>`. Average slope: `<slope>`%, elevation: `<elevation>` m. Soil properties: AWC = `<awc>`, Ksat = `<ksat>`, OM = `<om>`, BD = `<dbovendry>`, water content at 1/10 = `<wtenthbar>`, at 15 bar = `<wfifteenbar>`. |

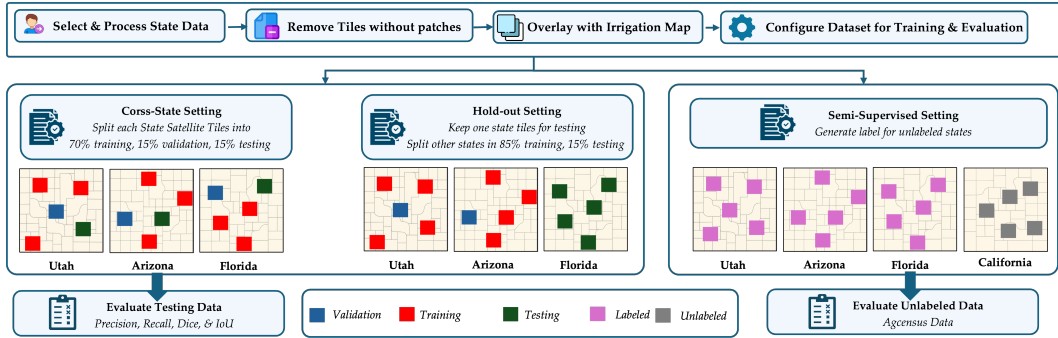

Figure S1: Caption

This integration embeds both surface-level observations and subsurface environmental priors into a consistent spatial framework and enables effective multimodal learning for irrigation classification.

## B.4 ML-ready Dataset Preparation

After processing and alignment, we extract a total of **1,484,301 ML-ready patches** across 20 states, with **293,890 labeled** and **1,190,206 unlabeled** examples. The labeled subset includes states such as *Washington, Utah, Colorado, Arizona, Florida*, and *Georgia* (some with partial coverage), while the unlabeled pool spans the remaining regions, enabling semi-supervised and pretraining applications. After reprojecting, masking, and patch extraction, we found **5,649 processed tiles** occupying **1,386.5 GB**, which represents a **77% reduction from raw data** in storage size compared to the original data footprint.

### B.4.1 Training and Evaluation Splits

We adopt two complementary strategies for splitting our dataset to support both intra-state generalization and cross-regional evaluation: a 70:15:15 spatial split within states and a leave-one-state-out protocol across states.

**Cross-State Splitting (70:15:15).** For each labeled state, we perform a split of the dataset into training (70%), validation (15%), and test (15%) sets. This split is performed at the tile level, where each tile represents a spatially contiguous region containing a dense grid of patches (typically covering a fixed Sentinel-2 footprint). Splitting at the tile level, as opposed to randomly splitting individual patches, ensures that data from the same geographic region does not appear in multiple splits—thereby preventing spatial leakage. This design is critical in remote sensing settings, where nearby patches are highly correlated due to shared land cover and climate conditions. Models trained and evaluated on spatially distinct regions are thus more likely to generalize to unseen geographies.

**Holdout-State Splitting (Leave-One-State-Out).** To evaluate out-of-distribution generalization, we perform a leave-one-state-out split. A target state is held out entirely for testing, while the remaining states contribute data to the training and validation sets. For the training states, we again split at the tile level and reserve a small portion (typically 10%) for validation. This setting simulates real-world scenarios where labeled data is unavailable in a new geographic region, and the model must transfer knowledge learned from other states. By maintaining strict spatial and regional disjointness, this split allows us to robustly measure a model's ability to generalize across distinct agricultural, climatic, and soil regimes.

### B.4.2 Irrigation Label Standardization

The irrigation annotations in our dataset originate from multiple geospatial sources and vary widely. Many raw labels encode specific equipment types (e.g., "Traveling Gun"), compound systems (e.g., "Center Pivot/Drip"), or non-standard notations ("Drip/None", "Unknown"). To unify this heterogeneous label space, we construct a manually curated mapping dictionary that normalizes all labels into three irrigation classes—*Drip*, *Sprinkler*, and *Flood*—along with fine-grained subclasses where available. We use a rule-based matching procedure over raw labels to assign each label to a class and a subclass. Labels that are ambiguous, mixed-mode, or uninformative (e.g., "None", "Unknown", "Center Pivot/Drip") are excluded to avoid introducing noise into the supervision signal. This label normalization step is essential for building a consistent and interpretable target space for training semantic segmentation and classification models. Table S3 summarizes the key mappings used.

## B.5 Impact of Raster Reprojection and Alignment

While this standardization facilitates multimodal alignment, it can introduce minor spatial smoothing or misalignment, particularly for coarser layers such as the USDA Cropland Data Layer (CDL) originally provided at 30 m resolution.

To quantify potential spatial distortions, we performed a reprojection consistency analysis on irrigation and crop masks. Each 30 m raster was resampled to the unified 10 m grid and then reconstructed back to 30 m. We compared the reconstructed rasters against the originals using overall accuracy and Cohen's $\kappa$ coefficient [61]. For irrigation masks, we obtained 97.6% accuracy and $\kappa = 0.96$ in Florida, and 93.9% accuracy with $\kappa = 0.89$ in Arizona. Crop masks yielded 93.4% accuracy and $\kappa = 0.92$ in Arizona, and 92.2% accuracy with $\kappa = 0.91$ in Colorado. Across all evaluated states, accuracy exceeded 90% and $\kappa$ values were above 0.85 on average.

These results indicate that the reprojection and resampling process preserves spatial integrity and label semantics with minimal information loss. Consequently, the uniform 10 m framework enables accurate alignment across modalities without degrading downstream model performance in the multimodal fusion pipeline.

Table S5: Complete List of Excluded Component Lists

**Soil Component**

10 to 20 inches deep over bedrock soils, 20 to 40 inches deep over bedrock soils, 20 to 60 inches deep over bedrock soils, 40 to 60 inches deep over bedrock soils, achimin, acid igneous rock land, acidic rock land, acidic rockland, aciidic rock land, active dune land, active land-slides - bench, active landslides - main scarp, aeric cryaquepts, ahl, airship, alcan, aldine, alice, alkaline soils, alluvium or colluvium, alo, alpin, alpineco, alumrock, amarose, amboycrater, andaquepts, ander, andic cryaquepts, andic cryochrepts, andic cryumbrepts, andic durixerepts, andic dystrocryepts, andic dystrocryrepts, andic dystroxerepts, andic dystrudepts, andic eu-trocryepts, andic eutrudepts, andic fragiumbrepts, andic humudepts, andic xerochrepts, andic xerumbrepts, andys, aned, ansel, antelope, antelope springs, antelope springs family, antelope springs variant, antero, aquandic cryaquepts, aquandic dystrocryepts, aquandic endoaquepts, aquandic halaquepts, aquandic humaquepts, aquandic humicryepts, aquariusmine, aquepts, aquic cryandepts, aquic cryumbrepts, aquic dystric xerochrepts, aquic dystrocryepts, aquic dystroxerepts, aquic dystrustepts, aquic ustochrepts, arave, archrock, areas of 10-20 percent slope, areas under water in, areas with 10-20 percent slope, areas with 30 to 55% slope, areas with 40-60 percent slope, areas with 50-100% slope, areas with 6-10 percent slope, aridic calciustepts, aridic calciustepts family, aridic haplustepts, aridic lithic haplustepts, ashnola, ashokawna, ashollow, atter, atter family, atwater, atwater variant, aut, ayar, badland, badlands, badrock, baileyboro, balman, barnetmine, barshaad, basalt rock land, baseline, basic rock land, batterson, battery, battlerock, baxter, bayboro, bedrock, beetlerock, bemjamin, benjamin, benlowe, bismarck, bisoodi, bitterwater, blackrock, blewett, bodiecreek, bonesteel, borrow, borrow pit, borrow pits, bottlerock, boulder, boulder lake, bouldercreek, boulderfan, boulder-jud, bouldery sandy alluvium, bouldery surface, brewster, bridgewater, bridgewater variant, briefly flooded areas, briefly flooded soils, britwater, broadwater, brocket, brockgulch, brock-gulch variant, brockliss, brocksburg, brockwell, budland, bunkwater, bushvalley, calciustepts, calcixerollic xerochrepts, calpet, calpine, calpine family, calpine variant, calwood, canisrocks, carmine, carshal, carshal family, carterlake, casrock, castee, castephen, castino extremely cobbly loam, castino gravelly silt loam, catherine, cavel, caverock, chain, chance, chaney, chatterdown, chazner, chemwash, chimrock, chinkmin, chirpchatter, choralmont, chumash, cinder quarries, clapper cobbly loam, clay pits, clayey shallow aridic ustorthents, clayman, cline, coal mine lands, coarser soils w/o sodium or salts, cobb, cobbly alluvial land, cobbly and gravelly soils, cobbly clay loam soils, cobbly soils, cobbly surface soils, colluvium deposits, colorock, contactmine, copalisrock, coppermine, cowmarsh, coyoterock, crestline, crockett, crop outcrop, crowther, cryandepts, cryaquepts, cryepts, cryochrepts, cryumbrepts, cumulic humaquepts, cumulic humixerepts, cushool

**Geomorphic Component**

aa lava flows, backswamps, barrier beaches, beach plains, beach ridges, beach terraces, beaches, blowouts, bogs, borrow pits, climbing dunes, complex landslides, dredge-deposit shoals, dune fields, dune lakes, dune slacks, dunes, estuarine tidal streams, falling dunes, filled marshlands, flood-tidal delta flats, flood-tidal delta slopes, flood-tidal deltas, foredunes, fringe-tidal marshes, gravel pits, interdunes, landslides, lava domes, lava fields, lava flows, lava plains, lava plateaus, lava trenches, longitudinal dunes, mangrove swamps, marshes, openpit mines, parabolic dunes, parna dunes, playa dunes, raised bogs, rock glaciers, rock pediments, rock spreads, rockfall avalanches, rockfalls, rotational landslides, salt marshes, sand pits, shoals, shrub-coppice dunes, sinkhole karst, sinkholes, slides, sloughs, spits, submerged upland tidal marshes, surface mines, swamps, tidal flats, tidal inlets, tidal marshes, urban land, wind-tidal flats

# C  Advancements Beyond Prior Works

IRRISIGHT represents a substantial advancement beyond our prior works, *IrrMap* and *KIIM*. Unlike *IrrMap*, which provided a small four-state dataset, and *KIIM*, which primarily introduced a model for modality fusion, IRRISIGHT (Table S7) establishes the first large-scale, multimodal, and vision–language–ready dataset for irrigation and agricultural water management. It integrates 37 structured

Table S6: Curated Soil and Geomorphic Features Deemed Unsuitable for Irrigation with Rationale

| Component | Curated Exclusion List | Rationale |
|---|---|---|
| **Soil** | sand dune, dune, gravel, cobbly, bouldery, channery, steep, slope, colluvium, badland | Poor water retention and unstable structure |
| | marsh, swamp, peat, muck, flood, water, wet, hydric, saline, sodic, tidal, playa, histosol | Saturated or saline soils that hinder rooting and aeration |
| | pit, mine, quarry, dumps, borrow | Artificial, excavated, or mined soils with low fertility or structure |
| | lava, rock, outcrop, bedrock | Hard or shallow substrates with limited rooting depth |
| | landslide, shoal | Geologically unstable land prone to movement or collapse |
| **Geomorphic** | dune, blowout, beach, slough, shoal | Shifting or erodible landforms with limited surface stability |
| | tidal, marsh, swamp, muck, peat, bog, wet, saline | Persistent wetland or tidal conditions incompatible with irrigation |
| | pit, mine | Excavated or mined geomorphic surfaces with disturbed profiles |
| | lava, rock | Volcanic or bedrock surfaces with low permeability |
| | landslide, sinkhole | Collapse-prone or unstable landforms |
| | urban | Impervious, artificial or urbanized surfaces |

geospatial and environmental features—including Sentinel-2 imagery, soil, hydrology, crop data, and deterministic text prompts—spanning 20 U.S. states and 1.48 million patches.

Beyond scale, IRRISIGHT introduces a scalable fusion pipeline, standardized multimodal schema, and model-agnostic cross-state benchmarks for evaluating both vision and vision–language architectures under spatial shift and class imbalance. These innovations transform prior single-purpose efforts into a unified and extensible resource for multimodal agricultural AI research.

## D    Experimentation Setting

## E    Dataset Benchmarking (Additional Details for Section 5)

Table S9 presents the mean and standard deviation of KIIM model confidence across four irrigation types for each of the 17 evaluated states. Non-irrigated patches consistently achieve the highest confidence scores (typically $>0.98$), indicating strong model certainty in identifying non-irrigated areas. In contrast, drip irrigation shows the lowest confidence overall (often below 0.50) with relatively high standard deviation, which suggests more ambiguity and spatial heterogeneity in its appearance. Flood and sprinkler irrigation classes show higher confidence values (compared to drip), with noticeable variation across states such as Arizona and Texas (see S2). Some states (e.g., Maryland, New Jersey) lack drip annotations, leading to missing values. The disparity in confidence reflects both inter-class difficulty and regional diversity in irrigation appearance. Overall, these results highlight where model predictions are most reliable and where further annotation

Table S7: Comparison of IRRISIGHT with prior works. '×' indicates not available.

| Aspect | IrrMap [43] | KIIM [27] | IRRISIGHT (This Work) |
|---|---|---|---|
| Core Contribution | Dataset (4 states) | Model (modality fusion) | Dataset + pipeline + benchmarks |
| States Covered | 4 | 4 (IrrMap subset) | 20 (6 labeled, 14 unlabeled) |
| Patch Count | ∼260K | ∼260K | 1.48M (293K labeled) |
| Drip Irrigation Samples | ∼18K | ∼18K | 61,723 patches with drip labels |
| Multimodal Data (Imagery + Soil + Water + Crop) | × | Limited (model input only) | ✓Full integration into dataset |
| Text Prompts (Soil, Geomorphology) | × | × | ✓Structured, rule-based, localized prompts |
| Hydrology (ET, GW, SW, Precipitation) | × | × | ✓Aligned per patch |
| Scalable Fusion Pipeline | × | × | ✓6.8 TB processed across formats |
| Cross-State Benchmarks | × | Model-specific (KIIM only) | ✓Model-agnostic LO-state-out evaluation |
| Vision–Language Models Evaluated | × | × | ✓CLIP, RemoteCLIP, BLIP-2 |
| Semi-Supervised Labeling (Synthetic) | × | × | ✓Confidence-filtered pseudo labels (Fig. 3) |

Table S8: Hyperparameter details for supervised experiments

| | Parameters |
|---|---|
| **Task Type** | Irrigation Type Segmentation |
| **Loss Function** | Cross Entropy |
| **Input Dimensions** | $224 \times 224$ |
| **Input Channels: RGB** | 3 |
| **Input Channels:RGB+LAND+CROP** | 24 |
| **Output Dimensions** | $224 \times 224$ |
| **Output Channels** | 4 |
| **Optimizer** | Adam |
| **Learning Rate** | $1.00 \times 10^{-2}$ |
| **No. Epochs** | 20 |
| **Model Selection** | Best (Dice on val.) |
| **Framework** | Pytorch Lightening |

or refinement is needed. Additionally, as an initial step, we identified two states (Nebraska and California) where verified irrigated land maps from past years (2005 and 2014) are available from the corresponding state's data [2]. By mapping our model-predicted irrigated areas with these maps, we observed accuracy scores of 64.5% in Nebraska and 64.6% in California. While limited in scope, this comparison provides a preliminary measure of model consistency with historical irrigation data.

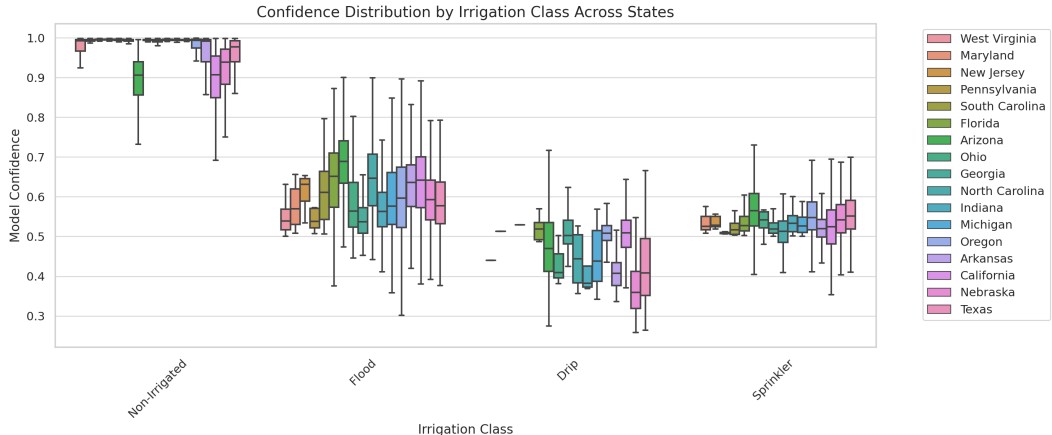

Figure S2: Confidence distribution of predicted label for 17 unlabeled states.

Table S9: Per-state confidence ($\mu \pm \sigma$) for synthetic labels across irrigation types.

| State | Drip | Flood | Sprinkler | Non-Irrigated |
|---|---|---|---|---|
| Arizona | $0.4747 \pm 0.0818$ | $0.6771 \pm 0.0935$ | $0.5687 \pm 0.0689$ | $0.8933 \pm 0.0598$ |
| Arkansas | $0.4113 \pm 0.0453$ | $0.6289 \pm 0.0672$ | $0.5130 \pm 0.0628$ | $0.9491 \pm 0.0780$ |
| California | $0.5043 \pm 0.0591$ | $0.6342 \pm 0.0879$ | $0.5211 \pm 0.0753$ | $0.8971 \pm 0.0681$ |
| Florida | $0.4982 \pm 0.0695$ | $0.6437 \pm 0.0927$ | $0.5343 \pm 0.0342$ | $0.9871 \pm 0.0269$ |
| Georgia | $0.5063 \pm 0.0504$ | $0.5455 \pm 0.0503$ | $0.5244 \pm 0.0220$ | $0.9903 \pm 0.0105$ |
| Indiana | $0.4165 \pm 0.0782$ | $0.5748 \pm 0.0634$ | $0.5332 \pm 0.0382$ | $0.9927 \pm 0.0097$ |
| Maryland | - | $0.5735 \pm 0.0504$ | $0.5362 \pm 0.0349$ | $0.9901 \pm 0.0158$ |
| Michigan | $0.4500 \pm 0.0887$ | $0.6004 \pm 0.0862$ | $0.5269 \pm 0.0429$ | $0.9928 \pm 0.0102$ |
| Nebraska | $0.3704 \pm 0.0672$ | $0.5931 \pm 0.0721$ | $0.5402 \pm 0.0648$ | $0.9198 \pm 0.0652$ |
| New Jersey | - | $0.6081 \pm 0.0508$ | $0.5352 \pm 0.0151$ | $0.9942 \pm 0.0060$ |
| North Carolina | $0.4428 \pm 0.0731$ | $0.6437 \pm 0.0875$ | $0.5151 \pm 0.0575$ | $0.9900 \pm 0.0260$ |
| Ohio | $0.4310 \pm 0.0635$ | $0.5879 \pm 0.0832$ | $0.5363 \pm 0.0297$ | $0.9928 \pm 0.0096$ |
| Oregon | $0.4984 \pm 0.0575$ | $0.5959 \pm 0.1010$ | $0.5481 \pm 0.0648$ | $0.9764 \pm 0.0364$ |
| Pennsylvania | - | $0.5660 \pm 0.0873$ | $0.5089 \pm 0.0039$ | $0.9942 \pm 0.0084$ |
| South Carolina | - | $0.6274 \pm 0.0918$ | $0.5229 \pm 0.0205$ | $0.9933 \pm 0.0115$ |
| Texas | $0.4212 \pm 0.0850$ | $0.5881 \pm 0.0738$ | $0.5512 \pm 0.0631$ | $0.9590 \pm 0.0441$ |
| West Virginia | - | $0.5499 \pm 0.0464$ | - | $0.9763 \pm 0.0306$ |

Table S10: Crop classification Macro-F1 across states using KIIM vs. CropScape. Higher is better.

| Model | AZ | CO | UT | WA | FL |
|---|---|---|---|---|---|
| KIIM | 57.7 | 84.5 | 61.2 | 88.9 | 70.0 |
| CropScape | 16.9 | 21.6 | 32.8 | 34.5 | 10.9 |

Table S11: Summary of Common Vegetation Indices, Their Purpose, and Use Cases

| Index | Formula | Purpose | Common Use Cases |
|---|---|---|---|
| NDVI (Normalized Difference Vegetation Index) | $\frac{(NIR-Red)}{(NIR+Red)}$ | Measures vegetation greenness and health | Crop monitoring, land cover classification, drought detection |
| EVI (Enhanced Vegetation Index) | $\frac{G \times (NIR-Red)}{(NIR+C_1 \times Red - C_2 \times Blue + L)}$ | Reduces atmospheric and soil background effects | More sensitive to high biomass regions |
| GNDVI (Green Normalized Difference Vegetation Index) | $\frac{(NIR-Green)}{(NIR+Green)}$ | More sensitive to chlorophyll content than NDVI | Water stress detection, photosynthetic activity |
| SAVI (Soil-Adjusted Vegetation Index) | $\frac{(NIR-Red)}{(NIR+Red+L)} \times (1+L)$ | Minimizes soil background influence | Vegetation monitoring in arid or semi-arid areas |
| MSAVI (Modified Soil-Adjusted Vegetation Index) | $\frac{2NIR+1-\sqrt{(2NIR+1)^2-8(NIR-Red)}}{2}$ | Further reduces soil influence compared to SAVI | Useful for sparse vegetation and dry land monitoring |
| RVI (Ratio Vegetation Index) | $\frac{NIR}{Red}$ | Alternative to NDVI, less sensitive to atmospheric conditions | Biomass and vegetation density analysis |
| CIgreen (Chlorophyll Index) | $\frac{NIR}{Green}-1$ | Estimates chlorophyll content | Plant health monitoring |
| NDWI (Normalized Difference Water Index) | $\frac{(NIR-SWIR)}{(NIR+SWIR)}$ | Measures water content in vegetation | Drought monitoring, irrigation management |
| PRI (Photochemical Reflectance Index) | $\frac{(Green-Blue)}{(Green+Blue)}$ | Measures plant stress and efficiency | Photosynthesis monitoring |
| OSAVI (Optimized Soil-Adjusted Vegetation Index) | $\frac{(NIR-Red)}{(NIR+Red+0.16)}$ | An improved version of SAVI that minimizes soil background effects while maintaining sensitivity to vegetation. | Used for vegetation monitoring in areas with moderate soil exposure. |
| WDRVI (Wide Dynamic Range Vegetation Index) | $\frac{a \times NIR - Red}{a \times NIR + Red}$ | A modified NDVI that enhances sensitivity to vegetation changes in high biomass areas. | Used in precision agriculture to track crop growth and stress detection. |
| NDTI (Normalized Difference Tillage Index) | $\frac{(SWIR1-SWIR2)}{(SWIR1+SWIR2)}$ | Differentiates between tilled and untilled soil, helping in soil disturbance and land management analysis. | Applied in soil erosion studies and land conservation planning. |