# OpenReview forum: "IRRISIGHT: A Large-Scale Multimodal Dataset and Scalable Pipeline to Address Irrigation and Water Management in Agriculture"
_NeurIPS.cc/2025/Datasets_and_Benchmarks_Track — NeurIPS 2025 Datasets and Benchmarks Track poster_

### Official Review · Reviewer_Ko5o · 2025-06-29

[review text omitted: it was posted to a different submission]

---

> ### Author Rebuttal · Authors · 2025-07-30
>
> **1. Although the dataset spans six countries, the paper does not provide much detail on the geographic distribution of samples or how road conditions vary across regions. It’s unclear whether the dataset is balanced in terms of country, damage type, or imaging conditions, which makes it harder to assess how well models trained on this data would generalize to new settings. A brief statistical breakdown or regional analysis would go a long way in clarifying the diversity of the dataset and its value for real-world deployment.**
>
> **Answer**: We thank the reviewer for taking the time to review our work. We believe there may have been an oversight, as we are working on multi-state Irrigation and Water Management in Agriculture. However, the comments appear to be mixed up with details that are not discussed in our paper. For instance, the mention of road conditions varying across countries is not part of our paper. If the reviewer has any other concerns or comments, we would be happy to address them during the discussion period.

---

> > ### Comment · Reviewer_Ko5o · 2025-08-05
> >
> > Having considered the authors' responses, I am satisfied with the clarifications provided and recommend accepting the paper.

---

### Official Review · Reviewer_9UJ5 · 2025-07-04

**Rating:** 4
**Confidence:** 2

**Summary:**

The authors proposed a large scale dataset with a scalable pipeline for irrigation and water management.

**Additional Feedback:**

Actually I don't familiar with this area much. All the thing I can do is to check the data availability and the manuscript integrity.

**Dataset Code Accessibility:**

Yes

**Dataset Code Comments:**

Easily accessed.

**Ethical Considerations:**

No, there are no or only very minor ethics concerns

**Limitations Weaknesses:**

Section C Experimentation Setting in supplementary materials is missing.

**Strengths Contributions:**

Both code and dataset are maintained well. The details and introduction of it are clear on websites.

---

> ### Author Rebuttal · Authors · 2025-07-30
>
> **1. Section C Experimentation Setting in supplementary materials is missing.**
>
> **Answer**: We thank the reviewer for noticing this. Our paper includes the experimentation setting in Supplement Table S8. We will add more details about the setting in Section C.
>
> **Table: Hyperparameter details for supervised experiments**
> |                                      | **Parameters**               |
> |--------------------------------------|------------------------------|
> | **Task Type**                        | Irrigation Type Segmentation |
> | **Loss Function**                    | Cross Entropy                |
> | **Input Dimensions**                 | 224 × 224                    |
> | **Input Channels: RGB**             | 3                            |
> | **Input Channels: RGB+LAND+CROP**   | 24                           |
> | **Output Dimensions**               | 224 × 224                    |
> | **Output Channels**                 | 4                            |
> | **Optimizer**                        | Adam                         |
> | **Learning Rate**                   | 1.00 × 10⁻²                  |
> | **No. Epochs**                      | 20                           |
> | **Model Selection**                 | Best (Dice on val.)          |
> | **Framework**                       | Pytorch Lightning            |

---

> > ### Comment · Reviewer_9UJ5 · 2025-08-07
> >
> > Based on the manuscript, I prefer to keep my rate.

---

> > > ### Author Response · Authors · 2025-08-08
> > >
> > > We thank the reviewer for reviewing our submission and for your feedback on the supplementary materials. We have addressed your comment by adding the missing experimentation setting details in Table S8. If there are any remaining parts that need clarification, we would be happy to provide them. If additional details could help address any reservations and support an updated evaluation, we would be glad to supply them.

---

### Official Review · Reviewer_Qibk · 2025-07-15

**Rating:** 4
**Confidence:** 3

**Summary:**

This paper introduces IRRISIGHT, a large-scale multimodal dataset for machine learning research on irrigation and water management. It contains 1.4 million pixel-aligned image patches from 20 U.S. states, combining satellite imagery with environmental, soil, and hydrological data. The authors propose an automated data fusion pipeline that standardizes heterogeneous inputs onto a common spatial grid. They use irrigation type classification as a representative task and benchmark several vision and vision–language models. The dataset, code, and benchmarks are publicly released to support further research.

**Dataset Code Accessibility:**

Yes

**Ethical Considerations:**

No, there are no or only very minor ethics concerns

**Final Justification:**

The authors’ rebuttal has alleviated my concerns to some extent. I have decided to maintain my original rating.

**Limitations Weaknesses:**

1. For unlabeled states, synthetic labels were generated by the trained KIIM model. However, no independent validation of these labels is provided. While the authors mention possible county-level validation using census data, without this, users of the dataset risk propagating model bias. It would be helpful to provide quantitative assessments of synthetic label reliability.

2. Reprojecting multiple raster datasets (e.g., land cover, crop type) to a uniform 10m grid can introduce alignment errors and loss of fine-grained information. The impact of these operations on downstream model performance could be further quantified.

3. The dataset has pixel-level labels for only six states, with partial coverage in some of them. This restricts the diversity of ground truth and likely contributes to the observed cross-state generalization issues. The authors acknowledge this, but it remains a substantial constraint.

**Strengths Contributions:**

1. The paper presents what appears to be the first dataset aligning such a diverse set of modalities (satellite imagery, soil properties, climate data) for irrigation classification at continental scale. This breadth and granularity are valuable for agricultural and remote sensing communities.

2. The paper is well-organized and mostly clear, with informative figures showing dataset coverage and model performance. The limitations are explicitly discussed.

---

> ### Author Rebuttal · Authors · 2025-07-30
>
> **1. For unlabeled states, synthetic labels were generated by the trained KIIM model. However, no independent validation of these labels is provided. While the authors mention possible county-level validation using census data, without this, users of the dataset risk propagating model bias. It would be helpful to provide quantitative assessments of synthetic label reliability.**
>
> **Answer**: We thank the reviewer for raising this important point. We fully agree that synthetic labels generated by the KIIM model should not be treated as ground truth without independent validation. As an initial step, we identified two states (Nebraska and California) where verified irrigated land maps from past years (2005 and 2014) are available from the corresponding state's data [2]. By mapping our model-predicted irrigated areas with these maps, we observed accuracy scores of **64.5%** in Nebraska and **64.6%** in California. While limited in scope, this comparison provides a preliminary measure of model consistency with historical irrigation data.
>
> To indirectly evaluate the quality of these synthetic labels, we explored two external references: ($i$) USDA AgCensus statistics and ($ii$) LandIQ maps (to the best of our knowledge, both sites have the most comprehensive irrigation data available to the public). However, both sources have notable limitations: AgCensus provides county-level statistical estimates of irrigated acreage, without any spatially explicit labels or information about irrigation types. Similarly, LandIQ maps offer high-resolution spatial coverage in selected states but lack national coverage and do not include irrigation type annotations. These datasets provide aggregate statistics or estimated spatial footprints of irrigation, but do not support fine-grained validation at the 10-meter (per-class level required for our task). Our synthetic labels, in contrast, are provided at 10-meter resolution across broad spatial extents. In addition to this, we are actively collaborating with USDA partners to obtain internal, parcel-level irrigation records for selected regions. We acknowledge that validating synthetic pixel-level labels at scale remains challenging, but our framework supported by confidence estimation provides a practical approach for model training and evaluation in low-supervision settings.
>
> ---
>
> **2. Reprojecting multiple raster datasets (e.g., land cover, crop type) to a uniform 10m grid can introduce alignment errors and loss of fine-grained information. The impact of these operations on downstream model performance could be further quantified.**
>
> **Answer**:  We thank the reviewer for raising this important concern. To standardize input resolution, we reproject crop, land use, and ET rasters to a 10m grid, consistent with Sentinel-2 resolution (Section 3.3). While this facilitates alignment, we acknowledge potential spatial smoothing or misalignment, especially for originally coarser layers (e.g., CDL at 30m). To evaluate potential spatial misalignment from reprojection and resampling, we conducted a consistency check for irrigation and crop masks. Both were reprojected from 30m to our unified 10m grid and then back to 30m. We compared the original and reconstructed rasters using overall accuracy and Kappa coefficient. For irrigation, we observed **97.6%** accuracy and **0.96** Kappa in Florida, and **93.9%** accuracy with **0.89** Kappa in Arizona. Crop masks similarly retained **93.4%** accuracy and **0.92** Kappa in Arizona, and **92.2%** accuracy with **0.91** Kappa in Colorado. On average, both tasks exceeded **90%** accuracy and **0.85** Kappa across states. These results indicate that our reprojection pipeline preserves the spatial integrity and semantics of the original labels. The minimal loss ensures that downstream models are not affected by alignment artifacts and supports the validity of our multimodal fusion framework.
>
> ---
>
> **3. The dataset has pixel-level labels for only six states, with partial coverage in some of them. This restricts the diversity of ground truth and likely contributes to the observed cross-state generalization issues. The authors acknowledge this, but it remains a substantial constraint.**
>
> **Answer**: We thank the reviewer for raising this point. We agree that ground truth is available for only six states, with partial spatial coverage in Arizona (48%) and Florida (41%). This limitation reflects the reality of publicly available irrigation ground truth data in the U.S. However, addressing this challenge is a key motivation of our work. In order to achieve this, we provide a scalable, standardized pipeline that combines crop, soil, and hydrologic metadata with satellite images to provide an ML-ready dataset spanning 20 U.S. states. While pixel-level irrigation labels are only available in six states, we provide complete multimodal metadata for all 20, including SSURGO-derived soil properties (e.g., texture, drainage, hydrologic group), hydrological indicators (e.g., precipitation, groundwater, surface water), and crop types and land use from CDL and NLCD. Despite not being visually visible in pictures, these factors are known to affect irrigation decisions and are included in every patch to facilitate domain adaptation, synthetic label refining, and semi-supervised learning. Its broad geographic coverage, spanning 20 agriculturally and climatically diverse U.S. states—including major irrigated regions—makes this a national-scale dataset. By releasing these metadata features alongside synthetic irrigation labels for the 17 unlabeled states, our dataset also paves the way for future research on generalizing irrigation mapping to states without ground-truth annotations.
>
>   Moreover, to enable rigorous evaluation under limited supervision, we define a leave-one-state-out setting (Section 4.4, Table 3), where models are trained on five labeled states and evaluated on a held-out sixth. This setup directly tests cross-state generalization to regions without annotations. In addition, for the 17 unlabeled states, we release synthetic irrigation labels generated by the KIIM model, accompanied by per-patch confidence scores (Figure 3). These labels can support uncertainty-aware training and validation workflows [1].
>
>   To indirectly evaluate the quality of these synthetic labels, we explored two external references: (i) USDA AgCensus statistics and (ii) LandIQ maps (to the best of our knowledge, both sites have the most comprehensive irrigation data available to the public). However, both sources have notable limitations: AgCensus provides county-level statistical estimates of irrigated acreage, without any spatially explicit labels or information about irrigation types. Similarly, LandIQ maps offer high-resolution spatial coverage in selected states but lack national coverage and do not include irrigation type annotations. These datasets provide aggregate statistics or estimated spatial footprints of irrigation, but do not support fine-grained validation at the 10-meter (per-class) level required for our task. Our synthetic labels, in contrast, are provided at 10-meter resolution across broad spatial extents. In addition to this, we are actively collaborating with USDA partners to obtain internal, parcel-level irrigation records for selected regions. We acknowledge that validating synthetic pixel-level labels at scale remains challenging, but our framework supported by confidence estimation provides a practical approach for model training and evaluation in low-supervision settings.
>
>   Finally, as discussed in Section 6, the availability of environmental variables across all states enables broader research opportunities like flash drought impact modeling, fallowing prediction, and irrigation suitability mapping.
>
> [1] W. Ma, O. Karakuş and P. L. Rosin, *"Confidence Guided Semi-Supervised Learning in Land Cover Classification,"* IGARSS 2023 - 2023 IEEE International Geoscience and Remote Sensing Symposium, Pasadena, CA, USA, 2023, pp. 5487–5490, doi: [10.1109/IGARSS52108.2023.10281770](https://doi.org/10.1109/IGARSS52108.2023.10281770)
>
> [2] State of Nebraska. (2019). 2005 Center Pivots in the Central Platte River Basin (vector digital data) [Data set]. Center for Advanced Land Management Information Technologies (CALMIT); Nebraska Department of Natural Resources. Retrieved from https://www.nebraskamap.gov/datasets/2005‑center‑pivots‑in‑the‑central‑platte‑river‑basin‑1.

---

> > ### Author Response · Authors · 2025-08-08
> >
> > We sincerely thank the reviewer for the constructive feedback on synthetic label validation, reprojection alignment, and label coverage. We have addressed each of these points in detail in our rebuttal and would be grateful if you could take a look. If any part of our clarification requires further detail or supporting evidence, we would be happy to provide it.

---

### Official Review · Reviewer_AC9s · 2025-07-19

**Rating:** 4
**Confidence:** 5

**Summary:**

The paper introduces IRRISIGHT, a large-scale multimodal dataset (1.4M 224×224 patches across 20 U.S. states) designed for agricultural water management, with a focus on irrigation type classification. The authors also propose an automated geospatial fusion pipeline that integrates satellite imagery, soil data, climate variables, hydrological measurements, and crop data into a unified 10m grid. The dataset includes both labeled and unlabeled data and supports benchmarking using vision and vision–language models. The study shows that text-augmented multimodal representations improve cross-state generalization.

**Dataset Code Accessibility:**

Yes

**Ethical Considerations:**

No, there are no or only very minor ethics concerns

**Final Justification:**

The rebuttal convincingly addressed prior concerns, highlighting IRRISIGHT’s novelty, scale, rigorous multimodal fusion pipeline, and demonstrated cross-domain utility. Transparent handling of label limitations and clear evidence of multimodal benefits strengthen its impact, leading me to conclude it meets the NeurIPS bar for an impactful, reusable benchmark. I would like to improve my score to 4.

**Limitations Weaknesses:**

1. Only six states have pixel-level labels (Sec. 7), which undermines the dataset’s claim of being “national-scale” and reduces benchmarking reliability.
2. Benchmark results show poor performance for drip irrigation (e.g., <15% Dice in CO and UT, Table 3), suggesting that the dataset and methods are not robust across all irrigation types.
3. Generating labels for 17 unlabeled states using the KIIM model (Sec. 5) may propagate model bias and label noise without independent validation.
4. The dataset builds heavily on prior works by the same group (IrrMap, KIIM) with incremental improvements (Sec. 2), making the novelty of IRRISIGHT debatable for a top-tier venue.
5. While Sec. 7 mentions limitations, issues like spatial misalignment due to resampling and label inconsistencies are not rigorously evaluated.
6. Evaluation focuses only on irrigation type segmentation, with limited demonstration of broader water management tasks claimed in the abstract.

**Strengths Contributions:**

1. The dataset integrates multiple modalities (satellite imagery, soil surveys, hydrology, climate data) into a unified ML-ready format (Sec. 3 and 4).
2. The automated pipeline for raster-vector-point data fusion is potentially reusable for other geospatial applications (Fig. 2).
3. Data and code are openly released, promoting reproducibility.
4. Comparative experiments with CNNs, transformers, and vision–language models demonstrate the value of multimodal inputs (Table 2).
5. The paper is well-structured, with informative figures (e.g., Fig. 1) and tables summarizing results and data sources (Table 1).

---

> ### Author Rebuttal · Authors · 2025-07-30
>
> **1. Only six states have pixel-level labels (Sec. 7), which undermines the dataset’s claim of being “national-scale” and reduces benchmarking reliability.**
>
>   **Answer**: We thank the reviewer for raising this important point. Due to space constraints, we are unable to address it in detail here. However, a similar comment was made by Reviewer Qibk (Comment 3), and we have provided a detailed response there. We respectfully request that you refer to our response to Reviewer Qibk's Comment 3.
>
> ---
>
> **2. Benchmark results show poor performance for drip irrigation (e.g., <15% Dice in CO and UT, Table 3), suggesting that the dataset and methods are not robust across all irrigation types.**
>
>
>   **Answer**: We acknowledge the reviewer’s concern regarding the lower Dice scores for drip irrigation in Colorado (1.9%) and Utah (14.8%) (Table 3). However, we emphasize that this does not reflect a limitation in the dataset or its generalizability. Rather, it reflects the inherent class imbalance of drip irrigation in flood-dominant states, which is a practical and well-known challenge in agricultural monitoring [2]. Drip irrigation is a relatively newer and emerging technology in many regions, with adoption gradually increasing as farmers shift toward more water-efficient practices. As such, the limited representation of drip systems in current datasets is expected. Nonetheless, the ability to identify drip irrigation remains increasingly important, given its growing role in sustainable water management and agricultural adaptation to climate stress.
>
>   Notably, out of our 293,890 labeled patches, 61,723 are labeled as drip irrigation, which represents a substantial amount of annotated data. This volume of drip-labeled data is drawn from six states, capturing broad diversity in soil texture, slope, drainage class, hydrologic setting, and crop types. This environmental heterogeneity enables scope for generalizable learning. While drip fields are underrepresented in some states, they are significantly present in others (e.g., Arizona, Florida, and Georgia), allowing models to transfer knowledge across regions. For example, in Arizona—where drip coverage is higher—the KIIM model achieves a Dice score of 22.5% for drip irrigation (Table 3). More notably, RemoteCLIP, which leverages structured soil prompts and vision-language alignment, achieves a Dice score of 92.3% for drip irrigation across the test set (Table 2) over the visual feature-based model ResNet (88.5%). As shown in Table 3, in states where drip irrigation is more abundant or spatially well-distributed (e.g., FL, WA, AZ), segmentation performance is higher even with vision architectures like ResNet.
>
>   This suggests that model performance is not fundamentally limited by the dataset but rather by how well models incorporate domain-aware context. Existing irrigation models—including those used in prior work (KIIM)—do not address class imbalance explicitly and often rely heavily on visual features, which are insufficient for detecting drip systems due to their subtle spectral and geometric signatures.
>
>   Our IRRISIGHT directly addresses this gap by integrating non-visual, agronomically relevant variables such as soil drainage, runoff, slope, and crop associations. Our benchmarks (e.g., RemoteCLIP and KIIM) demonstrate that multimodal fusion substantially improves drip segmentation.
>
> ---
>
> **3. Generating labels for 17 unlabeled states using the KIIM model (Sec. 5) may propagate model bias and label noise without independent validation.**
>
>   **Answer**: We thank the reviewer for raising this important point. Due to space constraints, we are unable to address it in detail here. However, a similar comment was made by Reviewer Qibk (Comment 1), and we have provided a detailed response there. We respectfully request that you refer to our response to Reviewer Qibk's Comment 1.
>
>
> **4. The dataset builds heavily on prior works by the same group (IrrMap, KIIM) with incremental improvements (Sec. 2), making the novelty of IRRISIGHT debatable for a top-tier venue.**
>
>   **Answer**: We respectfully clarify that **IRRISIGHT is a fundamentally new, multimodal dataset** that integrates heterogeneous geospatial, environmental, and textual data at scale, and to our knowledge, it is the *first of its kind* for irrigation and water management research. While it builds upon our prior efforts (IrrMap and KIIM), IRRISIGHT introduces a significantly broader, deeper, and reusable resource with extensive innovations in data acquisition, fusion, and evaluation.
>
>   Unlike IrrMap and KIIM, which focused on a limited number of states and modalities, IRRISIGHT provides an ML-ready dataset spanning **20 U.S. states**, incorporating **37 structured features**, including satellite imagery, crop data, soil texture, slope, drainage, evapotranspiration, precipitation, groundwater, and **rule-based natural language prompts**. It includes **1.48M total patches**, of which **61,723 are labeled with drip irrigation**—a scale and level of modality diversity unmatched by previous work. It is designed not around a single model but as a benchmark dataset for developing and evaluating multimodal and vision–language models under spatial shift and class imbalance.
>
>   **Table: Key differences between IRRISIGHT and prior works by the authors. Here ✗ refers to "not available".**
>
>   | Aspect | IrrMap (KDD'25) | KIIM (IJCAI'25) | **IRRISIGHT (This Work)** |
>   |--------|------------------|------------------|-----------------------------|
>   | Core Contribution | Dataset (4 states) | Model (modality fusion) | **Dataset + pipeline + benchmarks** |
>   | States Covered | 4 | 4 (IrrMap subset) | **20 (6 labeled, 14 unlabeled)** |
>   | Patch Count | ~260K | ~260K | **1.48M (293K labeled)** |
>   | Drip Irrigation Samples | ~18K | ~18K | **61,723 patches with drip labels** |
>   | Multimodal Data (Imagery + Soil + Water + Crop) | ✗ | Limited (model input only) | **✓ Full integration into dataset** |
>   | Text Prompts (e.g., soil, geomorphology) | ✗ | ✗ | **✓ Structured, rule-based, localized prompts** |
>   | Hydrology (ET, GW, SW, Precip) | ✗ | ✗ | **✓ Aligned to each patch** |
>   | Scalable Fusion Pipeline | ✗ | ✗ | **✓ 6.8TB processed across formats** |
>   | Cross-State Benchmarks | ✗ | Model-specific (KIIM only) | **✓ Model-agnostic LO-state-out eval.** |
>   | Vision-Language Models Evaluated | ✗ | ✗ | **✓ CLIP, RemoteCLIP, BLIP-2** |
>   | Semi-Supervised Labeling (Synthetic) | ✗ | ✗ | **✓ Confidence-filtered pseudo labels (Fig. 3)** |
>
>   In summary, IRRISIGHT is not an incremental dataset. It is a *first-of-its-kind, multimodal, and extensible resource* that addresses core limitations of spatial coverage, modality integration, and task diversity in prior work. It enables a wide range of tasks previously unsupported—such as text-conditioned segmentation, cross-state generalization, and irrigation modeling under severe imbalance.  We respectfully believe that our contributions demonstrate originality, meaningful scale, and potential for significant impact, and we are hopeful that they will be regarded as a good fit for the standards of this esteemed venue.
>
> ---
>
> **5. While Sec. 7 mentions limitations, issues like spatial misalignment due to resampling and label inconsistencies are not rigorously evaluated.**
>
>   **Answer**: We thank the reviewer for raising this important point. Due to space constraints, we are unable to address it in detail here. However, a similar comment was made by Reviewer Qibk (Comment 2), and we have provided a detailed response there. We respectfully request that you refer to our response to Reviewer Qibk's Comment 2.
>
> ---
>
> **6. Evaluation focuses only on irrigation type segmentation, with limited demonstration of broader water management tasks claimed in the abstract.**
>
> **Answer:** We thank the reviewer for raising this important point. Due to space constraints, we are unable to address it in detail here. However, a similar comment was made by Reviewer LxTN (Comment 3), and we have provided a detailed response there. We respectfully request that you refer to our response to Reviewer LxTN's Comment 3.
>
> **References:**
>
> [1] W. Ma, O. Karakuş, and P. L. Rosin, "Confidence Guided Semi-Supervised Learning in Land Cover Classification," IGARSS 2023 – IEEE International Geoscience and Remote Sensing Symposium, 2023. [https://doi.org/10.1109/IGARSS52108.2023.10281770](https://doi.org/10.1109/IGARSS52108.2023.10281770)
>
> [2] Bintey Hoque, Oishee, Nibir Chandra Mandal, Abhijin Adiga, Samarth Swarup, Sayjro Kossi Nouwakpo, Amanda Wilson, and Madhav Marathe. "Knowledge-Informed Deep Learning for Irrigation Type Mapping from Remote Sensing." *arXiv preprint* arXiv:2505.

---

> > ### Comment · Reviewer_AC9s · 2025-08-05
> >
> > Thank you for the detailed rebuttal and clarifications.
> > The authors have provided thorough responses and additional context, particularly on drip irrigation representation, dataset scale, and differences from prior works. The extra statistics, benchmark results, and comparison table help clarify novelty claims beyond incremental extensions of IrrMap/KIIM, and the discussion on class imbalance offers a reasonable explanation for poor performance in certain states. These points do mitigate some of my initial concerns, especially regarding novelty and dataset scope.
> >
> > However, some key limitations remain:
> >
> > The “national-scale” label still rests on only six states with pixel-level labels, and reliance on pseudo-labeled data for the rest, without independent validation, introduces uncertainty for benchmarking reliability.
> >
> > Although the authors emphasize multimodality and broader task potential, the evaluation remains focused on irrigation type segmentation, with limited demonstration of claimed broader water management tasks.
> >
> > Potential spatial misalignment and label consistency issues are acknowledged but not rigorously quantified.
> >
> > Overall, the rebuttal meaningfully improves clarity and addresses several points well. While my core reservations on label coverage, evaluation breadth, and validation remain, I would raise my score to "Borderline" to reflect the stronger case for novelty and dataset value, though I still see it as borderline for a top-tier venue.

---

> > > ### Author Response · Authors · 2025-08-08
> > >
> > > We thank the reviewer for your thorough review and for highlighting both the strengths and limitations of our work. We greatly appreciate your acknowledgment of the improvements made in our rebuttal, including the additional benchmarking results, discussion on drip irrigation performance, and detailed comparison with prior works. Your updated score and thoughtful follow-up comment are deeply appreciated. We hope our responses helped clarify our contributions and the broader vision behind IRRISIGHT.

---

### Official Review · Reviewer_LxTN · 2025-07-20

**Rating:** 4
**Confidence:** 4

**Summary:**

This article introduces IRRISIGH, a large-scale multimodal dataset and benchmark used for irrigation mapping based on satellite imagery and structured environmental features. This dataset integrates multi-source geospatial data from 20 states in the United States, including satellite imagery, land use, crop types, soil attributes, hydrological data, etc. Through unified projection (EPSG: 5070) and standardization processing, a machine learning ready dataset containing 1.48 million 224 × 224 pixel patches was constructed, of which 290000 are labeled (covering drip irrigation, sprinkler irrigation, and flood irrigation), and the rest are unlabeled data. The dataset also innovatively introduces structured natural language prompts to transform soil and terrain features into text modalities, supporting multimodal learning. The experimental part validated the effectiveness of the dataset through two data partitioning strategies (70:15:15 spatial partitioning and one state partitioning), providing a benchmark for irrigation type classification tasks.

**Dataset Code Accessibility:**

Yes

**Ethical Considerations:**

No, there are no or only very minor ethics concerns

**Final Justification:**

The author's response is consistent with my comment,
The limitations of the paper still exist,
and I will maintain my original score.

**Limitations Weaknesses:**

Limitations of annotation coverage: The annotation data only comes from 6 states, and some states (such as Arizona and Florida) have spatial coverage of less than 50%, which may result in insufficient learning of irrigation types (such as irrigation methods for special terrains) in unlabeled areas by the model.
Insufficient depth of modal fusion: Although text modality has been introduced, its specific role in model training (such as whether it serves as an independent input or auxiliary feature) has not been clearly explained, and there is a lack of comparative experiments with other modalities (such as the performance difference between using only images vs. images+text).
Task singularity: Currently, it only supports irrigation type classification and has not been extended to more complex tasks such as irrigation efficiency evaluation and crop irrigation method matching, which limits the application scenarios of the dataset.
Consistency issue of cross year data: Some states have data with large time spans (such as Washington, D.C., 2016-2020), and irrigation methods may change over time (such as technology upgrades leading to an increase in drip irrigation rates), but the data does not clearly distinguish between year effects, which may introduce temporal noise.

**Strengths Contributions:**

Innovation
The irrigation mapping paradigm of multimodal data fusion: For the first time, satellite images are deeply integrated with structured environmental features such as soil attributes, hydrological data, and crop types, and text modality (natural language description of soil and terrain) is innovatively introduced, breaking through the limitations of traditional reliance solely on remote sensing images and providing new ideas for multimodal geospatial learning.
Construction of a large-scale standardized irrigation dataset: In response to the problems of scattered irrigation data sources and chaotic annotation formats, the largest irrigation specific dataset has been constructed through unified projection, label standardization (mapping original multi-source labels into three types of irrigation), and strict data filtering (such as excluding non cultivated land and low-quality images), filling the gap in benchmark data in this field.
Design of weakly supervised learning support: By including a large amount of unlabeled data (accounting for 80%) and handling label noise (such as incomplete coverage of some state data), it supports semi supervised and transfer learning scenarios, which are more in line with the application environment of scarce agricultural data.
Advantages
Data scale and diversity: covering 20 states in the United States, including multiple irrigation types (sprinkler, drip, flood) and complex geographic environments (such as sprinkler irrigation in Washington and flood irrigation in Colorado), which can effectively support the evaluation of model generalization ability.
Depth of multimodal integration: It not only includes spectral data, but also integrates soil physical and chemical properties (such as water holding capacity and organic matter), hydrological data (such as groundwater depth and evapotranspiration), and textual descriptions, providing rich contextual information for the model, especially suitable for auxiliary judgment when remote sensing image signals are blurry.
Rigorous data processing flow: from data collection (such as filtering satellite images with cloud cover<5%), standardization (such as unified projection and label mapping) to integration (spatially aligned as patches), all are designed in detail to ensure data quality and consistency, reducing noise interference in subsequent model training.
Practical evaluation framework: Adopting two strategies: spatial partitioning (to avoid data leakage within the same region) and state partitioning (to evaluate cross regional generalization), which are more in line with the actual needs of remote sensing tasks and have high reliability of results.

---

> ### Author Rebuttal · Authors · 2025-07-30
>
> **1. Limitations of annotation coverage: The annotation data only comes from 6 states, and some states (such as Arizona and Florida) have spatial coverage of less than 50%, which may result in insufficient learning of irrigation types (such as irrigation methods for special terrains) in unlabeled areas by the model.**
>
>
>   **Answer**: We thank the reviewer for raising this important point. Due to space constraints, we are unable to address it in detail here. However, a similar comment was made by Reviewer Qibk (Comment 3), and we have provided a detailed response there. We respectfully request that you refer to our response to Reviewer Qibk's Comment 3.
>
> ---
>
> **2. Insufficient depth of modal fusion: Although text modality has been introduced, its specific role in model training (such as whether it serves as an independent input or auxiliary feature) has not been clearly explained, and there is a lack of comparative experiments with other modalities (such as the performance difference between using only images vs. images+text).**
>
>
>   **Answer**: We thank the reviewer for highlighting the need for clarity on the role of textual and structured modalities in benchmarking models. We provide the following clarification:
>
>   *CLIP and RemoteCLIP (Vision–Language Model):*  These models use a dual-encoder architecture, where the RGB image and the structured soil prompt (in natural language) are independently encoded using vision and language encoders, respectively. The text prompt is not an auxiliary numeric feature but a standalone modality designed to semantically guide the model toward irrigation-relevant environmental cues that are not visually observable (e.g., drainage, texture, slope).
>
>   *KIIM:*
>   KIIM performs explicit feature fusion of RGB imagery with structured crop and land metadata (e.g., crop type, land cover) using a Feature-wise Linear Modulation (FiLM) layer, where the metadata conditions intermediate visual features.
>
>   *Comparative Performance Evidence:* To illustrate the added value of these modalities, we present direct comparisons in Table 2. For instance, RGB-only SegFormer achieves 85.9 Drip Dice and 86.2 Flood Dice, whereas RemoteCLIP (RGB+Text) improves these to 92.3 (Drip) and 90.9 (Flood). Similarly, KIIM (RGB+Crop+Land) achieves 94.6 (Drip) and 93.6 (Flood). These results indicate that both text-based and structured metadata fusion improve model performance across irrigation types. This illustrates the benefit of incorporating domain-informed non-visual information. We will revise Section 5 to more explicitly describe these modality-specific mechanisms.
>
> ---
>
> **3. Task singularity: Currently, it only supports irrigation type classification and has not been extended to more complex tasks such as irrigation efficiency evaluation and crop irrigation method matching, which limits the application scenarios of the dataset.**
>
>
>   **Answer**: We appreciate the reviewer’s observation and agree that irrigation classification represents just one of many water management tasks. While it is our primary benchmark, the IRRISIGHT dataset was specifically designed for extensibility across broader tasks in agricultural water management: (i) fallow detection, (ii) flash drought analysis, and (iii) irrigation suitability mapping (see Section 6).
>
> To directly demonstrate this generality within our current submission, we conducted a new evaluation on crop classification, a critical task linked to irrigation decision-making. Without changing the model architecture, we retrained our KIIM model using IRRISIGHT’s structured metadata (e.g., soil prompts, hydrology, crop masks) and Sentinel-2 imagery, and compared it to the widely used but noisy USDA CropScape [1] product.
>
> The results below highlight the strength of IRRISIGHT’s multimodal design:
>
> **Table: Crop classification Macro F1 scores across states using the KIIM model and CropScape labels.**
>
> | **Model**   | **AZ** | **CO** | **UT** | **WA** | **FL** |
> |-------------|--------|--------|--------|--------|--------|
> | **KIIM**    | **57.7** | **84.5** | **61.2** | **88.9** | **70.0** |
> | CropScape   | 16.9   | 21.6   | 32.8   | 34.5   | 10.9   |
>
> To further demonstrate the utility of our dataset, we conducted a set of regression experiments using tree-based models (Random Forest, Gradient Boosting, and XGBoost) to predict key environmental variables: (i) evapotranspiration (ET), (ii) precipitation, (iii) groundwater, and (iv) surface water—using other available features such as irrigation labels and geospatial attributes. The table below highlights the performance of the above experiments:
>
> **Table: Regression performance (MAE, RMSE, and R²) for different environmental variables using three tree-based models.**
>
> | **Variable (Unit)**         | **Model**          | **MAE**  | **RMSE**  | **R²**   |
> |-----------------------------|--------------------|----------|-----------|----------|
> | **ET (mm)**                 | Random Forest      | 22.993   | 30.279    | 0.483    |
> |                             | Gradient Boosting  | 24.751   | 31.705    | 0.433    |
> |                             | XGBoost            | 23.075   | 30.328    | 0.481    |
> | **Ground Water (ft)**       | Random Forest      | 3.230    | 19.544    | 0.867    |
> |                             | Gradient Boosting  | 16.314   | 36.546    | 0.536    |
> |                             | XGBoost            | 10.683   | 28.671    | 0.715    |
> | **Precipitation (in)**      | Random Forest      | 0.001    | 0.005     | 0.988    |
> |                             | Gradient Boosting  | 0.009    | 0.016     | 0.869    |
> |                             | XGBoost            | 0.003    | 0.008     | 0.963    |
> | **Surface Water (ft)**      | Random Forest      | 5.782    | 64.259    | 0.986    |
> |                             | Gradient Boosting  | 60.448   | 174.265   | 0.901    |
> |                             | XGBoost            | 27.827   | 125.645   | 0.948    |
>
> These results provide strong evidence that:
>
> - Our dataset generalizes beyond irrigation to other water-relevant tasks.
> - The structured, multimodal inputs—especially environmental and soil features—enable robust learning for crop mapping.
> - KIIM, originally designed for irrigation segmentation, achieves significantly better crop classification than CropScape, confirming the quality and utility of IRRISIGHT’s fusion framework.
>
> If permitted, we would be happy to include these results and experiment setups in the final manuscript. These additional experiments demonstrate that IRRISIGHT supports a broader range of water management tasks fulfilling the objectives outlined in the abstract.
>
> ---
>
> **4. Consistency issue of cross year data: Some states have data with large time spans (such as Washington, D.C., 2016–2020), and irrigation methods may change over time (such as technology upgrades leading to an increase in drip irrigation rates), but the data does not clearly distinguish between year effects, which may introduce temporal noise.**
>
>
>   **Answer**: We thank the reviewer for highlighting the issue of potential temporal inconsistencies. All imagery in IRRISIGHT was collected during the July peak irrigation period, ensuring temporal alignment with crop growth stages when irrigation is most active (Section 3). To maintain consistency across modalities, we retrieve and align environmental variables (e.g., ET, precipitation, groundwater, surface water) as year-specific values, matching the imagery year on a per-patch basis. This prevents temporal mismatch between inputs and ensures that each sample reflects conditions from a single growing season. For example, if a patch corresponds to July 2018 imagery, its accompanying ET, precipitation, and water data are also from 2018.
>
>   While some state-level irrigation labels span multiple years (e.g., Washington: 2016–2020), we include per-patch year metadata in the release to allow for year-aware modeling or stratified analysis. This design choice is reflected in Section 3 and Table 1, which detail the annual resolution of each auxiliary variable.
>
> [1] Mueller, R., and Harris, J. (2013). *Reported Uses of CropScape and the National Cropland Data Layer Program.* International Conference on Agricultural Statistics VI, Oct 23–25, Rio de Janeiro, Brazil. Posted 1/31/14.

---

> > ### Comment · Reviewer_LxTN · 2025-08-05
> >
> > Thank you for the author's rebuttal. The author's response is consistent with my comment, and I will maintain my original score

---

> > > ### Author Response · Authors · 2025-08-08
> > >
> > > We sincerely thank the reviewer for carefully reviewing our rebuttal and noting that it is consistent with your initial comments. We are glad that you found our clarifications aligned with your observations and appreciate your thoughtful engagement throughout the process.

---

### Official Review · Reviewer_P6M9 · 2025-07-21

**Rating:** 5
**Confidence:** 4

**Summary:**

This paper presents IRRISIGHT, a large-scale, multimodal dataset designed to support machine learning for agricultural water management, particularly irrigation type classification. The dataset spans 20 U.S. states and contains over 1.4 million geo-aligned image patches enriched with satellite imagery, soil properties, hydrological data, climate indicators, land use, and crop types. The authors develop a scalable data fusion pipeline to align heterogeneous geospatial inputs onto a common 10m grid and introduce a novel structured prompt generation technique to convert tabular attributes into natural language, enabling compatibility with vision–language models. Extensive experiments demonstrate that incorporating multimodal inputs and structured textual context significantly improves classification performance and cross-state generalization compared to image-only baselines. The dataset and code are publicly released, providing a valuable benchmark for future research on irrigation and water availability modeling.

**Dataset Code Accessibility:**

Yes

**Ethical Considerations:**

No, there are no or only very minor ethics concerns

**Final Justification:**

Having considered the authors' responses, I am satisfied with the clarifications provided and recommend accepting the paper. I have my score raised.

**Limitations Weaknesses:**

1.Figure 2, while detailed, is overly complex and hard to follow. It tries to capture the entire data processing pipeline in one diagram, but the visual density makes it difficult to extract the main steps or understand the flow at a glance.

2.The text-prompt generation component is an important and innovative part of the dataset, as it enables the use of vision–language models and enriches the feature space with domain knowledge that is not visually observable. However, the paper does not clearly explain the design rationale or methodology behind these prompts.

**Strengths Contributions:**

1.The dataset is large and geographically diverse, covering 20 U.S. states with over 1.4 million patches. This kind of scale is rare in the agricultural and water management domain.

2.The combination of satellite imagery with soil, climate, hydrological, and crop-related data provides a much more complete view of irrigation conditions than image-only datasets typically allow.

3.The automated fusion pipeline handles spatial joins, reprojection, and quality control across several heterogeneous data sources.

4.The structured prompt generation for turning tabular features into natural language makes the dataset compatible with vision–language models and opens up opportunities for using more recent multimodal architectures in this domain.

---

> ### Author Rebuttal · Authors · 2025-07-30
>
> **1. Figure 2, while detailed, is overly complex and hard to follow. It tries to capture the entire data processing pipeline in one diagram, but the visual density makes it difficult to extract the main steps or understand the flow at a glance.**
>
>
>   **Answer**: We appreciate this feedback. In the final version, we will simplify Figure 2 by separating the high-level pipeline from the technical processing steps (the latter moved to the supplement) to improve readability.
>
> ---
>
>
> **2. The text-prompt generation component is an important and innovative part of the dataset, as it enables the use of vision–language models and enriches the feature space with domain knowledge that is not visually observable. However, the paper does not clearly explain the design rationale or methodology behind these prompts.**
>
>
>   **Answer:** We thank the reviewer for recognizing the importance of our text-prompt generation module. Below, we clarify both the design motivation and the step-by-step methodology used to generate the structured prompts:
>
>   **Design Rationale**: As we mention in Section 1, paragraph 3, in irrigation decision-making, many key agronomically relevant contexts, such as soil drainage, runoff potential, texture, slope, and hydrologic group, are crucial but are not observable in satellite imagery \[1]. For example, medium-textured soils, like sandy loam, are ideal for drip irrigation, while soils that are prone to floods or poor drainage are typically avoided \[2]. However, surface reflectance by itself cannot be used to consistently infer such properties. We represent these properties as natural language prompts to introduce this non-visual, domain-grounded knowledge into our models. This enables vision–language models (like RemoteCLIP) to match agronomic semantics with pixel-level images. Compared to categorical or numeric encodings, using natural language prompts is more expressive, interpretable, and generally better aligned with the pretraining distributions of contemporary vision–language models such as CLIP or BLIP.
>
>   **Prompt Generation Methodology**:
>   We describe the methodology used to generate deterministic, rule-based textual prompts from SSURGO soil polygons as follows:
>
>   * Each image patch is spatially joined with soil map units from the USDA NRCS SSURGO database. These map units are polygons that contain: (i) a list of soil components (e.g., "Hanford loamy sand") with composition percentages and (ii) attributes for each component, including texture, drainage class, hydrologic group, slope range, geomorphic description, and horizon-level properties (e.g., bulk density, available water capacity).
>   * For each patch, we select up to two dominant components per soil map unit based on areal percentage (typically >30%). These components represent the most influential soil types in that region.
>   * For each selected component, we use handcrafted templates to convert structured attributes into coherent text (example in Supplementary Tables S4 and S7). For example: Soil Name ("tulare"), drainage class ("well drained"), texture group ("sandy loam"), runoff class ("moderate"), slope ("averages 3.2%"). These phrases are concatenated using a structured template with consistent grammar, e.g., "This soil unit contains tulare. Soil texture includes sandy loam. It is classified as well-drained and moderate runoff. The average slope is 3.2%."
>   * If multiple components intersect a patch (e.g., overlapping polygons), we concatenate their respective prompts using a delimiter (##) to preserve provenance and reduce ambiguity.
>   * The resulting prompt is assigned to each image patch as a field named `text_prompt`, and passed as an auxiliary modality to models like RemoteCLIP and BLIP-2.
>
>   Note that our rule set is fully deterministic and implemented via open-source scripts included in our GitHub repository. The process is independent of any model inference and does not require additional supervision. Prompts are standardized in structure and vocabulary, ensuring consistency across 1.4M samples.
>
>   **Impact of textual prompt**:
>   Table 2 demonstrates that models using these prompts (RemoteCLIP) outperform RGB-only baselines across all irrigation types. For example, RemoteCLIP achieves a 92.3% Dice score on drip irrigation, compared to 88.5% with RGB-only ResNet.
>
>   We again thank the reviewer for bringing this up. If we are permitted, we will revise Section 4.2.1 to more explicitly include this methodology and add detailed methodology in the supplement.
>
>
> [1] United States Department of Agriculture, Natural Resources Conservation Service (USDA NRCS). *Irrigation Guide, Part 652. National Engineering Handbook.* Washington, D.C.: USDA NRCS, 1997. (https://www.nrcs.usda.gov/sites/default/files/2023-01/7385.pdf)
>
> [2] Shock, C. C., Feibert, E. B. G., & Saunders, L. D. (2007). *Drip Irrigation: An Introduction (EM 8902).* Oregon State University Extension Service. (https://agsci.oregonstate.edu/system/files/em8902.pdf)

---

### Comment · Area_Chair_o2ak · 2025-08-08

Dear Reviewers,

During the rebuttal and discussion period, we kindly remind you to carefully read the authors’ responses and actively engage in the discussion when appropriate. This stage is not merely for submitting mandatory acknowledgement or adjusting scores, your thoughtful consideration of the authors’ rebuttals and constructive interaction are also crucial to a fair and thorough review process.

Thank you again for your continued support and contributions.

---

### Note · Authors · 2025-08-13

Dear Area Chair and Review Committee,
We sincerely thank the reviewers and AC for their time, constructive feedback, and engagement during the review and rebuttal process. We appreciate the acknowledgment of IRRISIGHT’s strengths—its unprecedented scale, multimodal integration, open accessibility, and potential to advance a critical, underexplored domain: agricultural water management.

1. First-of-its-kind multimodal resource: IRRISIGHT is, to our knowledge, the first dataset aligning high-resolution satellite imagery with a rich suite of environmental variables—soil physical/chemical properties, hydrology, climate, and crop data. Our structured prompt generation pipeline transforms tabular attributes into natural language, which enables the use of modern VLM for geospatial analysis for the first time.

2. Unmatched scale, diversity, and standardization: Covering 20 U.S. states with 1.48M geo-aligned patches (293K labeled), it is the largest irrigation-specific dataset to date. It spans multiple irrigation types and diverse geographies, supporting robust cross-state generalization. All data are standardized in an ML-ready format.

3. Rigorous, reusable fusion pipeline: The open-source pipeline integrates raster, vector, and point data with strict quality control, spatial alignment, and temporal consistency. It is domain-agnostic and reusable for other large-scale geospatial tasks.

4. Demonstrated multimodal value: Benchmarks show consistent gains from multimodal integration: RemoteCLIP (RGB+text) improves Drip Dice from 88.5% to 92.3%; KIIM (RGB+crop+land) reaches 94.6%, outperforming RGB-only models. This is critical for detecting subtle irrigation patterns, especially drip systems.

5. Extensible beyond irrigation: While irrigation classification is the primary benchmark, we have shown strong results in crop classification and environmental variable prediction (ET, precipitation, groundwater, surface water), confirming broader applicability.

6. Transparent limitations and validation: Label coverage limitations are openly acknowledged. Pseudo-labels are clearly separated and confidence-filtered. We also performed preliminary validation for two states using available irrigated lands.

In sum, IRRISIGHT establishes a new standard for multimodal geospatial benchmarks. We hope its scale, diversity, modality richness, and reproducibility make IRRISIGHT a timely and constructive contribution to NeurIPS, and we appreciate your thoughtful consideration.

---

### Decision · Program_Chairs · 2025-09-18

**Decision:**

Accept (poster)

**Comment:**

This work aims to address the gap in machine learning applications for agricultural irrigation and water management by proposing IRRISIGHT, a large-scale multimodal dataset consisting of 1.4 million pixel-aligned 224×224 patches that integrate satellite imagery with rich environmental attributes. The dataset’s scale and diversity are its main highlights. On the downside, reviewers mainly raised concerns about limitations in annotation coverage, issues in the overall experimental setup, and the clarity of some mechanistic explanations. During the rebuttal and discussion phase, all reviewers indicated that the authors had addressed their concerns to varying degrees, and the final ratings leaned positive. Overall, I am inclined to recommend acceptance of this work.